

# Ship emissions measurement in the Arctic from plume intercepts of the Canadian Coast Guard *Amundsen* icebreaker from the *Polar 6* aircraft platform

A. A. Aliabadi[1], J. L. Thomas[2], A. Herber[3], R. M. Staebler[1], W. R. Leaitch[4], K. S. Law[2], L. Marelle[2,5], J. Burkart[6], M. Willis[6], J. P. D. Abbatt[6], H. Bozem[7], P. Hoor[7], F. Köllner[7], J. Schneider[8], and M. Levasseur[9]

[1]Processes Research Section, Air Quality Research Division, Atmospheric Science and Technology, Science and Technology Branch, Environment Canada, Toronto, Canada
[2]Sorbonne Universités, UPMC Univ. Paris 06, Université Versailles St-Quentin, CNRS/INSU, LATMOS-IPSL, Paris, France
[3]Alfred Wegener Institute - Helmholtz Center for Polar and Marine Research, Bremerhaven, Germany
[4]Climate Chemistry Measurements and Research Section, Climate Research Division, Atmospheric Science and Technology, Science and Technology Branch, Environment Canada, Toronto, Canada
[5]TOTAL S.A, Direction Scientifique, Tour Michelet, 92069 Paris La Defanse, France
[6]Department of Chemistry, University of Toronto, Canada
[7]Institute for Atmospheric Physics, Johannes Gutenberg University of Mainz, Germany
[8]Particle Chemistry Department, Max Planck Institute for Chemistry, Germany
[9]Department of Biology, Université Laval, Quebec, Canada

*Correspondence to:* A. A. Aliabadi (Aliabadi@aaa-scientists.com)

**Abstract.** Decreasing sea ice and increasing marine navigability in northern latitudes have changed Arctic ship traffic patterns in recent years and are predicted to increase annual ship traffic in the Arctic in the future. Development of effective regulations to manage environmental impacts of shipping requires an understanding of ship emissions and atmospheric processing in the Arctic environment.

As part of the summer 2014 NETCARE (Network on Climate and Aerosols) campaign, the plume dispersion and gas and particle emission factors of emissions originating from the Canadian Coast Guard Amundsen icebreaker operating near Resolute Bay, NU, Canada have been investigated. The Amundsen burnt distillate fuel with 1.5 wt % sulfur. Emissions were studied via plume intercepts using aircraft measurements, an analytical plume dispersion model, and using the FLEXPART-WRF

Lagrangian particle dispersion model. The first plume intercepts by research aircraft were carried out on 19 July 2014 during the operation of the Amundsen in the open water. The second and third plume intercept measurements were carried out on 20 and 21 July 2014 when the Amundsen had reached the ice edge and operated under icebreaking conditions. Typical of Arctic marine navigation, the engine load was low compared to cruising conditions for all of the plume intercepts. The mea-

sured species included mixing ratios of $CO_2$, $NO_x$, CO, $SO_2$, particle number concentration (CN), refractory Black Carbon (rBC), and Cloud Condensation Nuclei (CCN). The results were compared to similar experimental studies in mid latitudes.





Plume expansion rates ($\gamma$) were calculated using the analytical model and found to be $\gamma = 0.75 \pm 0.80$, $0.93 \pm 0.37$, and $1.19 \pm 0.39$ for plumes 1, 2, and 3, respectively. These rates are smaller than

prior studies conducted at mid latitudes, likely due to polar boundary layer dynamics, including reduced turbulent mixing compared to mid latitudes. All emission factors were in agreement with prior observations at low engine loads in mid latitudes. Icebreaking increased the $NO_x$ emission factor from $EF_{NO_x} = 22.3 \pm 8.0$ to $57.8 \pm 11.0$ and $65.8 \pm 4.0$ g kg $-$ diesel$^{-1}$ for plumes 1, 2, and 3, likely due to change in combustion temperatures. The CO emission factor was $EF_{CO} = 6.4 \pm 11.7$,

$6.8 \pm 2.2$ and $5.0 \pm 1.0$ g kg $-$ diesel$^{-1}$ for plumes 1, 2, and 3. The rBC emission factor was $EF_{rBC} = 0.20 \pm 0.04$ and $0.25 \pm 0.12$ g kg $-$ diesel$^{-1}$ for plumes 1 and 2. The CN emission factor was reduced while icebreaking from $EF_{CPC} = 1.96 \pm 0.41$ to $0.43 \pm 0.11$ and $0.47 \pm 0.04 \times 10^{16}$ kg $-$ diesel$^{-1}$ for plumes 1, 2, and 3. At 0.6 % supersaturation, the CCN emission factor was lower than observations in mid latitudes at low engine loads with $EF_{CCN} = 1.63 \pm 0.41$ to $1.06 \pm 0.32$ and $0.28 \pm 0.07 \times$

$10^{14}$ kg $-$ diesel$^{-1}$ for plumes 1, 2, and 3.

## 1   Introduction

International shipping is responsible for approximately 3.3 % of global $CO_2$ emissions, 5 to 8 % of global anthropogenic $SO_2$ emissions, and 2 % of global Black Carbon (BC) emissions (Lack and Corbett, 2012). The regulations for air pollutants released by ships are set by the International Con-

vention for the Prevention of Pollution from Ships (MARPOL) within the International Maritime Organization (IMO) accessible at http://www.imo.org/en/OurWork/Environment/PollutionPrevention/ Pages/Default.aspx. In addition, specific sensitive regions are subject to more stringent limits for Emissions Control Areas (ECAs), such as those in effect for the Baltic Sea, the Mediterranean Sea, and the Caribbean Region. In the high Arctic, including Canadian waters, there is currently no ECA

established, despite the very sensitive nature of the Arctic environment and ecosystems. At the same time, the decreasing sea ice and increasing marine navigability in the shipping season have already increased annual traffic in the Canadian Arctic in the recent decades (Pizzolato et al., 2014). Future projections in Arctic ship traffic also suggest increased emissions by mid century (Corbett et al., 2010a; Winther et al., 2014). Development of effective regulations require an understanding of ob-

served ship emissions and processing in the Arctic environment.

Ship emissions measurements from land-based, marine-based, and airborne platforms have been reported in numerous studies (e.g. von Glasow et al., 2003; Chen et al., 2005; Agrawal et al., 2008; Petzold et al., 2008; Lack et al., 2009; Williams et al., 2009; Petzold et al., 2010; Lack et al., 2011; Petzold et al., 2011; Berg et al., 2012; Khan et al., 2012a; Lack and Corbett, 2012; McLaren et al.,

2012; Alföldy et al., 2013; Diesch et al., 2013; Eckhardt et al., 2013; Buffaloe et al., 2014; Cappa et al., 2014; Kivekäs et al., 2014; Balzani Lööv et al., 2014; Pirjola et al., 2014; Aliabadi et al., 2015c; Beecken et al., 2015; Marelle et al., 2015; Roiger et al., 2015). However, studies that attempt to



measure ship emissions in the Arctic from land, marine, and airborne platforms are limited (Eckhardt et al., 2013; Aliabadi et al., 2015c; Marelle et al., 2015; Roiger et al., 2015).

The sniffer method including plume intercepts above background values is commonly used to study ship emissions factors, where the increase in concentration of pollutants compared to the background atmosphere can be observed (Berg et al., 2012; Balzani Lööv et al., 2014; Pirjola et al., 2014; Beecken et al., 2015). One prior ship plume intercept study (described in Roiger et al., 2015; Marelle et al., 2015) has been performed in the European Arctic during summer, when ships operate in the

open water (no sea ice operations). Due to the particular physical and chemical properties of the Arctic boundary layer, it is important to study ships operating in sea-ice and other Arctic conditions in order to compare ship emission and plume processing to results from studies at mid latitudes. Differences in background concentrations of reactive species in the atmosphere between high and mid latitudes, including gases and aerosols, may result in substantially different processing of ship pol-

lutants in the Arctic. Furthermore, ship conditions when breaking and conducting operations within sea ice are different, including partial engine load setting for speed reduction and ice breaking that could affect the emissions factors for pollutants significantly (e.g. Lack and Corbett, 2012).

von Glasow et al. (2003) and Petzold et al. (2008) have used a power law relationship to model ship plume dispersion as a growing semi ellipse within the marine boundary layer. The plume growth

rate has been successfully estimated for various ships in mid latitudes and found to be in a similar range. Here, we use the same method to estimate plume expansion rates for the Arctic boundary layer. The cold and statically stable marine boundary layer in the Arctic, which is governed by effects of surrounding ice and small changes in solar zenith angle, is likely to impact dispersion and expansion of the ship plumes differently (Anderson and Neff, 2008; Aliabadi et al., 2015a, b). Here,

we use the same power law model in the Arctic and compare the predicted plume expansion with prior studies using this method in the mid latitudes, with different boundary layer dynamics.

Many parameters change ship emissions factors including engine load, fuel type, and emissions abatement technologies. Ship speed reduction results in better fuel economy and lower $CO_2$ emissions, due to reduced drag on the ship hulls (Jalkanen et al., 2012; Lack and Corbett, 2012). It also

reduces particulate matter, BC, and $NO_x$ emissions factors in addition to reducing particulate matter size (Agrawal et al., 2008; Khan et al., 2012b; Petzold et al., 2010, 2011; Cappa et al., 2014). On the other hand, operating ship engines at partial load increases Organic Carbon (OC), BC, and CO emissions factors (Agrawal et al., 2008; Petzold et al., 2011; Jalkanen et al., 2012; Khan et al., 2012b; Lack and Corbett, 2012; Cappa et al., 2014).

Sulfur in ship fuels is primarily converted to $SO_2$ gas, increasing particle emissions by forming secondary sulphates (e.g. Jalkanen et al., 2012; Lack and Corbett, 2012). Lower sulfur content in ship fuels reduces particulate matter and BC emission factors (Lack et al., 2011; Petzold et al., 2011; Alföldy et al., 2013), particle size (Lack et al., 2011), and modifies the concentration of aerosols that serve as Cloud Condensation Nuclei (CCN) (e.g. Petzold et al., 2010).



Slide valves, water-in-fuel emulsion, diesel particulate filters, emulsified fuel, and sea water scrub-
bing are key abatement technologies to reduce emissions factors for various pollutants (Corbett et al.,
2010b; Lack and Corbett, 2012). While effective in reducing emissions factors for certain species,
these technologies cannot reduce all emissions factors simultaneously. Some remedies result in re-
duced fuel economy (higher $CO_2$ emissions) due to running auxiliary pumps and other equipment,
while others reduce some emissions factors at the expense of increasing the others (Corbett et al.,
2010b; Miola et al., 2010).

### 1.1    Research objectives

In this study we use measurements from airborne plume intercepts to estimate emissions factors
for the Amundsen ship, while operating in the Arctic and burning low sulfur fuel, for gaseous and
particle pollutants. In addition, we study the geometrical evolution of the Amundsen's plume in
the Arctic marine boundary layer. We compare these observations to other similar studies in mid
latitudes. The first plume measurement was carried out on 19 July 2014 during the operation of the
CCGS Amundsen in the Lancaster Sound of the Northwest Passage ($74°,18'$ N, $83°,54'$ W). The
second and third plume measurements were carried out on 20 and 21 July 2014 after the CCGS
Amundsen reached the ice edge and operated under ice conditions, north of Somerset Island, less
than 50 km from Resolute Bay. These measurements provide differences in plume characteristics
between operation under open water conditions as well as sea ice conditions in the Arctic.

## 2    Methods

### 2.1    Specifications of Amundsen icebreaker

The Amundsen (IMO: 7510846) belongs to the Canadian Coast Guard fleet with full specifications
available at http://www.ccg-gcc.gc.ca/Fleet/Vessel?vessel_id=3. It is an Arctic Class 3 vessel, 98.2 m
long, with gross tonnage of 5911.0 t, and maximum speed of 16.0 kts. The propulsion is provided
by a diesel electric AC/DC system with 6 main Alco M251F engines of total power 13200 kW. It
has 3 Alco MLW251F generators and a Caterpillar 398 emergency generator. During the campaign,
Amundsen burned marine distillate fuel that contained 1.5 wt % sulfur content (ISO 8217 2010 DMA
Fuel Standard).

### 2.2    Airborne measurements

The airborne instrument platform was the Polar 6 aircraft, a DC-3 converted to a Basler BT-67,
owned and operated by the German Alfred Wegener Institute - Helmholtz Center for Polar and Ma-
rine Research (Fig. 1) (Leaitch et al., 2015). Below, experimental methodologies for the measure-
ments of state parameters and meteorology, gas phase, and particle phase pollutants are presented.



### 2.2.1 State parameters and meteorological measurements

State parameter and meteorological measurements are performed by an AIMMS-20 instrument, manufactured by Aventech Research Inc., Barrie, Ontario, Canada. The instrument consists of three modules. The Air Data Probe (ADP) measures the three-dimensional, aircraft-relative flow vector (true air speed, angle-of-attack, and sideslip). The temperature and relative humidity sensors are located in the aft section of the probe for protection. A three-axis accelerometer pack facilitates direct turbulence measurement. The Inertial Measurement Unit (IMU) consists of three gyros and three accelerometers providing the aircraft angular rate and acceleration. A GPS module provides the aircraft 3D position and inertial velocity. Horizontal and vertical wind speeds are measured with accuracies of 0.50 and $0.75 \, \mathrm{m \, s^{-1}}$, respectively. The accuracy and resolution for temperature measurement are 0.30 and $0.01 \, ^\circ \mathrm{C}$. The accuracy and resolution for relative humidity measurement are 2.0 and 0.1 %. The sampling frequency is greater than 40 Hz, but in this study a sampling frequency of 1 Hz is used.

### 2.2.2 Gas phase measurements

Trace gas $CO_2$ measurement was based on infrared absorption using a LI-7200 enclosed $CO_2/H_2O$ Analyzer from LI-COR Biosciences GmbH. In-situ calibrations during the flight were performed on a regular time interval of 15 to 30 min using a NIST traceable calibration gas with a known $CO_2$ concentration at atmospheric levels. The uncertainty for the measurement of $CO_2$ is 0.3 ppmv relative to the standard. Trace gas CO was measured with an Aerolaser ultra fast carbon monoxide (CO) monitor model AL 5002 based on VUV-fluorimetry. The same in-situ calibrations during inflight were performed. The calibrations and zero measurements allowed for corrections of instrument drifts increasing the stability and accuracy of the instrument, thus leading to an uncertainty of $\pm 2.3$ ppbv relative to the standard.

Trace gas $NO_x$ measurement was based on chemiluminescence using a Thermo Scientific 42$i$ $NO-NO_2-NO_x$ analyzer with a time resolution of 1 s and an uncertainty of 0.4 ppbv. Trace gas $SO_2$ measurement was based on UV Fluorescence light-scattering using a Thermo Scientific Model 43$i$-TLE Enhanced Trace Level $SO_2$ analyzer with a time resolution of 1 s and an uncertainty of 1 % of reading or 0.2 ppbv, whichever is greater. Trace gas $O_3$ measurement was based on UV photometry using a Thermo Scientific 49$i$ analyzer with a time resolution of 10 s and an uncertainty of 0.2 ppbv. For simplicity trace gas mixing ratio units of [ppbv] is presented as [ppb] hereafter.

### 2.2.3 Particle phase measurements

Particle number concentrations greater than 5 nm diameter were measured with a TSI 3787 water-based ultrafine Condensation Particle Counter (CPC), sampling at a flow rate of $0.6 \, \mathrm{L \, min^{-1}}$ and a time resolution of 1 s. These measurements are referred to as CPC hereafter.





Aerosol particle size distributions from 70 nm to 1 µm were measured by a Droplet Measurement Technology (DMT) Ultra High Sensitivity Aerosol Spectrometer (UHSAS) that uses scattering of 1054 nm laser light to detect particles (Cai et al., 2008). The time resolution was 1 s and the measurements are referred to as UHSAS hereafter.

     Cloud Condensation Nuclei (CCN) concentrations were measured by a DMT CCN Model 100
counter operating behind a DMT low pressure inlet at a reduced pressure of approximately 650 hPa and a nominal water supersaturation of 1 %. The effective supersaturation at 650 hPa was determined to be approximately 0.6 % and was held constant throughout the study to allow for more stability of measurements, improved response, and to examine the hygroscopicity of smaller particles. The time resolution was 1 s and the measurements are referred to as CCN hereafter.

Extensive calibrations and evaluations for CPC, UHSAS, and CCN measurements were performed in the laboratory prior to integration of the instruments on the aircraft and again with instrumentation in the aircraft at Resolute Bay. Full discussions can be found in the study by Leaitch et al. (2015).

     Particle size distribution for particle diameters greater than 0.25 µm was measured using a Sky Optical Particle Counter (OPC model 1.129). Measurements were based on 90° scattering light and
a time resolution of 6 s. The accuracy is ±3 % at 1 sigma confidence. These measurements are referred to as OPC hereafter.

     The refractory black Carbon (rBC) was measured using a single particle soot photometer (SP2) from DMT Boulder. The SP2 (Schwarz et al., 2010) is an instrument able to evaluate individual aerosol particles for the rBC mass content, size and mixing state based on the laser-induced incan-
descence method and can gather information on the scattering part of the aerosol ensemble. The time resolution was 1 s and the measurements are referred to as SP2 hereafter.

     Particle sampling is described in full detail by Leaitch et al. (2015) and was performed so that the efficiency of particle transmission to instruments would be close to 100 % for particles from 20 nm to 1 µm in diameter.

**2.2.4   Power law model for plume growth**

The methodology of von Glasow et al. (2003) describes plume dispersion with a power law which models plume dimensions in horizontal ($w_{pl}$) and vertical ($h_{pl}$) directions.

$$\begin{cases} w_{pl}(t) = w_0 \left( \frac{t}{t_0} \right)^{\alpha} \\ h_{pl}(t) = h_0 \left( \frac{t}{t_0} \right)^{\beta} \end{cases} \tag{1}$$

with $w_0$ and $h_0$ being plume dimensions at reference time ($t_0 = 1$ s) and $\alpha$ and $\beta$ being plume ex-
pansion rates in the horizontal and vertical directions. Fitted values for expansion rates are provided in the literature for mid latitude marine boundary layers (von Glasow et al., 2003; Petzold et al., 2008); however, it remains to be verified if expansion rates are similar or different over the Arctic marine boundary layer. The power law describes plume cross-section with a semi-elliptic shape with



area $A_{pl} = \frac{\pi}{8} w_{pl} h_{pl}$. It is assumed that plume expansion in the vertical direction is inhibited when
it reaches the top of the marine boundary layer, where subsequent expansion only continues in the
horizontal direction.

A convenient and practical way to fit for plume expansion rates is to intercept a portion of the
plume and measure the mixing ratio of a chemically inert species in the plume such as $CO_2$. Assum-
ing uniform dilution of such species in the plume, it is possible to derive a relationship between the
species mixing ratio in the plume ($c_{pl}$) and expansion rate coefficients ($\alpha$ and $\beta$),

$$ln\left(c_{pl}(t) - c_{bgd}\right) = -\gamma ln\left(\frac{t}{t_0}\right) + ln\left(c_{pl}(t_0) - c_{bgd}\right) \tag{2}$$

where $c_{bgd}$ is the background mixing ratio of the species and $\gamma$ is either $\alpha + \beta$ for plumes not reaching
marine boundary layer or $\alpha$ for plumes that evolve after reaching the top of the marine boundary
layer. Then $\gamma$ is the expansion rate and $m = -\gamma$ is the slope of the linear relationship. The reference
time for this calculation is independent from the reference time introduced earlier. Since mixing in
real plumes is not uniform, time or cross sectional-averaging of the airborne-measured mixing ratio
and multiple measurements at various distances from the source are necessary to arrive at a better
estimate for the plume expansion rate.

### 2.2.5 Estimation of plume age

Plume age can be estimated by the aircraft measurements. For this, plume intercepts are first mapped
on a latitude/longitude plot. This provides a scatter plot to which a plume center line is fitted with
a high order polynomial. The wind measurements on board of the aircraft closest to each point
on the center line are then used to estimate wind velocity along the fitted plume center line. This
methodology enables plume age estimation at each intercept by calculating the time it takes for the
plume center line to reach the nearest location to the intercept using the following formula

$$T(L) = \int\limits_{l=0}^{l=L} \frac{dl}{U(l)} \tag{3}$$

which is a line integral starting from the plume center line origin ($l = 0$) to the nearest plume inter-
cept on the center line ($l = L$). $U(l)$ is the estimated horizontal wind speed along the plume center
line.

### 2.2.6 Emissions factors per kilogram of fuel burnt

A common method to calculate emission factors (EF) in $[\text{g kg} - \text{diesel}^{-1}]$ is the net peak area
method (Alföldy et al., 2013) using the $CO_2$ balance concept (Hobbs et al., 2000). For a pollutant
measurement in units of [ppb], the molecular weights of carbon and a gaseous pollutant species of





interest are considered. Given the carbon mass percent in diesel fuel ($87 \pm 1.5\,\%$; (Cooper, 2005)), the emissions factor for species X can be expressed as,

$$\text{EF}_\text{X}[\text{g kg}-\text{diesel}^{-1}] = \frac{C(\text{X})[\text{ppb s}]}{C(\text{CO}_2)[\text{ppb s}]} \times \frac{MW_\text{X}[\text{g mol}^{-1}]}{MW_\text{C} = 12[\text{g}_\text{C}\,\text{mol}^{-1}]} \times 0.87[\text{g}_\text{C}\,\text{g}_\text{diesel}^{-1}] \times 1000[\text{g kg}^{-1}]$$
(4)

where $C()$ represents the mixing ratio of species above background levels integrated over time for an entire peak and $MW$ stands for molecular weight, which for carbon is 12. EF can be estimated at a reference customary plume age or as an average for all plume encounters.

For pollutant measurement in units of mass concentration (e.g. [$\mu$g m$^{-3}$]), EF can be estimated using the same methodology, however, the molecular weight of the pollutant is not necessarily needed since the measurement in units of mass per volume is already available (Lack et al., 2009),

$$\text{EF}_\text{X}[\text{g kg}-\text{diesel}^{-1}] = \frac{C(\text{X})[\mu\text{g m}^{-3}\,\text{s}]}{C(\text{CO}_2)[\text{ppb s}]} \times 1620[\text{g }\mu\text{g}^{-1}\,\text{m}^3\,\text{ppb kg}-\text{diesel}^{-1}]$$
(5)

where the constant $1620\,[\text{g }\mu\text{g}^{-1}\,\text{m}^3\,\text{ppb kg}-\text{diesel}^{-1}]$ accounts for the same carbon mass percent in diesel fuel. For particle emissions in units of [cm$^{-3}$], the emissions factor can be calculated using (Lack et al., 2009),

$$\text{EF}_\text{X}[\text{kg}-\text{diesel}^{-1}] = \frac{C(\text{X})[\text{cm}^{-3}\,\text{s}]}{C(\text{CO}_2)[\text{ppb s}]} \times 1.62 \times 10^{15}[\text{cm}^3\,\text{ppb kg}-\text{diesel}^{-1}]$$
(6)

where the constant $1.62 \times 10^{15}\,[\text{cm}^3\,\text{ppb kg}-\text{diesel}^{-1}]$ accounts for the same carbon mass percent in diesel fuel.

If a modal emissions factor with units of [g kWh$^{-1}$] is reported, which applies to both gaseous and particle phases, it is possible to convert it to units of [g kg$-$diesel$^{-1}$] if emissions factor for $CO_2$ is also available in units of [g kWh$^{-1}$]. The conversion is provided by,

$$\text{EF}_\text{X}[\text{g kg}-\text{diesel}^{-1}] = \frac{\text{EF}_\text{X}[\text{g kWh}^{-1}]}{\text{EF}_{\text{CO2}}[\text{g kWh}^{-1}]} \times \frac{MW_{\text{CO}_2} = 44[\text{g}_{\text{CO}_2}\,\text{mol}^{-1}]}{MW_\text{C} = 12[\text{g}_\text{C}\,\text{mol}^{-1}]} \times 0.87[\text{g}_\text{C}\,\text{g}_\text{diesel}^{-1}] \times 1000[\text{g kg}^{-1}]$$
(7)

Similarly, if a modal emissions factor with units of [kWh$^{-1}$] is reported, which applies to number of particles, it is possible to convert it to units of [kg$-$diesel$^{-1}$] if the emissions factor for $CO_2$ is also available in units of [g kWh$^{-1}$]. The conversion is provided by,

$$\text{EF}_\text{X}[\text{kg}-\text{diesel}^{-1}] = \frac{\text{EF}_\text{X}[\text{kWh}^{-1}]}{\text{EF}_{\text{CO2}}[\text{g kWh}^{-1}]} \times \frac{MW_{\text{CO}_2} = 44[\text{g}_{\text{CO}_2}\,\text{mol}^{-1}]}{MW_\text{C} = 12[\text{g}_\text{C}\,\text{mol}^{-1}]} \times 0.87[\text{g}_\text{C}\,\text{g}_\text{diesel}^{-1}] \times 1000[\text{g kg}^{-1}]$$



$$(8)$$

The calculated EF for conserved pollutants, such as $CO_2$, is constant and not a function of plume age. However, for other pollutants it may increase (production) or decrease (consumption) as a function of

plume age. Due to limited number of plume intercepts in this study, we compute average emissions factors for all plume intercepts.

### 2.3 FLEXPART-WRF plume dispersion modeling

In order to study, the dispersion of ship emissions in the Polar boundary layer, we use the FLEXPART-WRF model (Brioude et al. (2013), website: flexpart.eu/wiki/FpLimitedareaWrf), a Lagrangian par-

ticle dispersion model based on FLEXPART (Stohl et al., 2005). FLEXPART-WRF is driven by meteorology from the Weather Research and Forecasting (WRF) Model (Skamarock et al., 2005), with the specifics of the WRF run for NETCARE provided in Wentworth et al. (2015). Here we ran FLEXPART-WRF in forward mode to study plume dispersion from the Amundsen. Running FLEXPART-WRF in forward mode is useful for studying the specific plume structure and emissions

location for the case of a single moving point source (e.g. a single ship) involving complex movements (moving ship location with time) within a complex and changing meteorological situation. FLEXPART-WRF was run in forward mode using the known ship location. Particles were released each minute along the ship track using a source extending 100 m vertically and horizontally centered on the ship location, from 17 July 2014 00:00 UTC to 22 July 2014 00:00 UTC. An arbitrary emis-

sions source strength was assumed for the model run (mass of particles emitted) and considered to be constant in time for the duration of the run. FLEXPART-WRF output was saved on a grid approximately 1 km × 1 km (resolution of 0.01 ° Latitude × 0.05 ° Longitude) in order to obtain results on a similar spatial scale as the plume sampling.

## 3   Results and discussion

### 3.1   Meteorological context

Plume intercepts in the three consecutive days are referred to as plume 1 (19 July 2014), plume 2 (20 July 2014), and plume 3 (21 July 2014) studies. The flights were planned in advance using WRF and FLEXPART-WRF forecasts (not shown) so that the aircraft could efficiently sample ship emissions downwind of the stack. Following the campaign, WRF was run using ECMWF (European Centre

for Medium-Range Weather Forecasts) analysis as initial and boundary conditions, (see Table 2 of Wentworth et al., 2015), in order to refine forecast meteorology and to interpret campaign data. The quality of the WRF predicted meteorology has been evaluated using measurements made on-board both the research aircraft and ship, indicating the forecast accurately predicts the meteorological situation during plume sampling (flight tracks shown in Figure 2). Surface wind speed and wind




direction predicted by WRF during plume sampling are shown in Figure 3. During the first plume
sampling on 19 July 2014, the flight was conducted west of the ship location due to the easterly winds
throughout the plume sampling, characterized by high wind speeds above 10 m s$^{-1}$ (Figure 2a) in
Lancaster sound (Figure 3a). For the second plume, on 20 July 2015, the ship was located just north
of Somerset Island (Figure 3b) and the flight sampled ship emissions southwest of the ship, between

the ship and the Somerset Island. The meteorological situation near the flight was less consistent
in the measurement region on 20 July 2014, indicated by the variable wind directions and lower
wind speeds measured (Figure 2b). This is also shown by the variable wind speeds and directions
in the region of the flight in Figure 3b. On 21 July northwesterly winds throughout Lancaster sound
resulted in plume sampling to the southeast of the ship, with consistent wind speeds (but lower than

on 19 July 2014) during the plume sampling (Figure 2c).

We also characterize boundary layer dynamics using balloon soundings launched from the ship at
the times of the flights for plumes 2 and 3 (Figure 4). For plume 1, there was no balloon sounding,
therefore we show only the WRF model results for comparison. The measurements and the model
are in good agreement, noting that the model under predicted wind speeds below 100 m on 21 July

2014 compared to the measurements. This is also seen in the flight track on 21 July 2014 (Figure
2c). We also, however, note that WRF model does perform better than the ECMWF analysis (wind
speed and wind direction) for this flight. The boundary layer is statically stable and the boundary
layer height is calculated from measurements to be 387 m and 177 m for plume 2 and 3 studies,
respectively, using the method of bulk Richardson number developed by Mahrt (1981) and later

used by Aliabadi et al. (2015a). Vertical gradients in potential temperature and wind speed show that
the emissions are predicted to be mixed into a shallow boundary layer on all three days, both during
operations in the open water (plume 1) and within sea ice (plumes 2 and 3).

### 3.2   Ship operating conditions

It is known that both ship speed and engine load influence total fuel burned and emission factors.

For the Amundsen, ship speed is not directly correlated with engine load for two reasons. First,
the Amundsen operates on a diesel-electric system, which could provide propulsion power using
electricity while the engines are off or operating at partial load. Second, because of the specifics of
ships operating in the Arctic within sea ice, even during stationary conditions, the engine may be
running to power ice breaking operations. The average ship speed during plume 1, 2, and 3 studies

were 3.23±0.25 kts, 1.31±1.92 kts, and 0.09±0.30 kts, respectively. The variation in ship speed is
calculated using one standard deviation, noting that both plume 2 and 3 studies involved ice breaking.

### 3.3   FLEXPART-WRF ship plume modeling

In order to show the relationship between the emissions from the ship (plumes) on different days
and the flight pattern, we use FLEXPART-WRF partial columns and vertical cross sections. Given





the low boundary layer heights, maps of the plume distributions were calculated by summing the mass of particles in the lowest 350 meters above the ocean/sea ice. Three example partial columns during plume sampling are shown in Figure 5. The corresponding locations of plume crossings along the flight tracks, derived from measured peak enhancements in $NO_x$ (see section 3.4), which are used later for emissions factor calculations (see section 3.7), are shown in Figure 6. The two

figures indicate that the plume intercepts are in the same locations as the partial columns predicted by FLEXPART-WRF. This agreement provides confidence that the measured enhancements in trace gases and aerosols originate from the ship emissions.

The predicted vertical distribution of emissions along and across the plumes are shown in Figure 7. The model indicates that the ship emissions are predicted to be below 300 meters when the ship

was operating in the open water (plume 1) and are predicted to stay in the lowest 100 meters when the ship was operating in sea ice (plumes 2 and 3). The vertical cross sections across the plumes (Figure 7 panels marked e - *across plume*) show that the horizontal distribution of emissions is predicted several kilometers across during plume crossing. It is also worth noting that all of the plume crossings used for emissions factor calculation and analysis were below 90 m (Figure 6), corresponding to the

most concentrated portion of the ship plume as predicted by FLEXPART-WRF (Figure 7). The exact properties of the ship plumes are determined by the combination of the meteorological conditions, emissions injection location (horizontal and vertical extent), and the ship movements. This analysis also shows that the predicted plumes mix slowly with the background air in the strongly stable Arctic boundary layer, with implications for the fate of emissions and plume processing.

**3.4  Ship plume pollutant identification**

Plume intercepts have been identified using the methodology of Petzold et al. (2008) where a statistically significant change in mixing ratio of a non-decaying gaseous pollutant with respect to background has been observed. Figure 8 shows an example time series where pollution peaks in the plume are evident. This time series is used to identify the location and timing of ship plume crossings

(shown in Figure 6), which is also referred to as an excess or peak event. To identify plume crossings the $NO_x$ mixing ratio with a threshold of 2 ppb has been used, which was preferred over $CO_2$ due to unpredictable background variations in the $CO_2$ mixing ratio. In this method, the background for $NO_x$ mixing ratio was computed by averaging three consecutive measurements before and after the threshold. Once time stamps for $NO_x$ peak events were identified, all other pollutant peaks were

identified using these times, without the need for a threshold. A time shift between peak events was expected between the reference instrument ($NO_x$) and any other instrument since they sampled air at different locations on the sampling line. This shift was identified and corrected by maximizing the coefficient of determination ($R^2$) for the 1-1 mixing ratio plots between a pair of instruments.



### 3.5 Analytical model of ship plume expansion

Using airborne meteorological and $CO_2$ mixing ratio measurements, the power law plume expansion model (eq. 1 and e.q. 2), and the estimated plume age (eq. 3), the plume geometrical evolution can be explained for all of the plumes. In this calculation, the vertical variations in wind speed and direction are accounted for. Using the methodology in section 2.2.5 a plume age could be assigned to $n = 6$ data points for plume 1, $n = 7$ data points for plume 2, and $n = 18$ data points for plume

3 studies. Figure 9 shows the measured expansion rates for these data points. The expansion rate is the magnitude of slope for the fitted lines ($\gamma = \alpha + \beta$), and it is calculated as $\gamma = 0.75 \pm 0.80$, $0.93 \pm 0.37$, and $1.19 \pm 0.39$ for plumes 1, 2, and 3, respectively. The uncertainty is computed for the regression analysis as one standard deviation. This compares reasonably well to values reported in the literature for mid latitudes. Petzold et al. (2008) find $\gamma = 1.5 \pm 0.06$ for a ship plume expansion in

the English Channel, and von Glasow et al. (2003) find a best guess value of $\gamma = 1.35$ for a number of previous studies also in mid latitudes. Our lower expansion rate suggests that ship plumes in the Arctic marine boundary layer mix with the background to a lesser extent compared to mid latitude due to the statically stable conditions.

### 3.6 Changes in gas mixing ratios and particle concentrations

#### 3.6.1 Gas Pollutants

Figure 10 shows the scatter plot for excess gas pollutants versus excess carbon dioxide. In all three instances, excess carbon monoxide and nitrogen dioxide correlate with excess carbon dioxide, while ozone titration accounts for a negative correlation between excess ozone and excess carbon dioxide. Given the lower detection limit of the instrument, there was no trace of $SO_2$ in the plume as measured

by the aircraft. It has been verified by $SO_2$ measurements on-board of the ship that the mixing ratios were below $2\,\mathrm{ppb}$, indicating that the exhaust after treatment on the Amundsen effectively removes this species (Wentzell, 2015, Personal Communication).

  Table 1 shows the results for linear regression analysis for excess gas pollutants versus excess carbon dioxide. The only insignificant coefficient of determination belongs to excess carbon monoxide

in plume 1. The regression slope ($b$) for excess oxides of nitrogen in plume 1 is a factor of 2 less than plumes 2 and 3, attributed to ice-breaking conditions, and hence higher engine temperature (but not necessarily engine load), during plume 2 and 3 studies.

#### 3.6.2 Particle Pollutants

Figure 11 shows the scatter plot for excess particle concentrations versus excess carbon dioxide. A

correlation is noticeable for all instruments. The SP2 instrument was not functional during plume 3 study. Table 2 shows the results for linear regression analysis for excess particle concentrations versus excess carbon dioxide. The regression slope ($b$) for plume 1 associated with CPC, OPC,





UHSAS, and CCN concentrations are factors of 5, 4-10, 2-3, 2-5 higher than plumes 2 and 3. This may be related to possible higher engine load (also vessel speed), but lower engine temperature according to section 3.6.1, for this plume.

### 3.7 Emissions factors

Emissions factors (EF) in the literature are reported in different ways. Some studies report EF for one ship or a fleet of ships operating under various engine loading conditions or fuel types (Petzold et al., 2008; Lack et al., 2011; Khan et al., 2012a; Alföldy et al., 2013). Another common approach is to group EF based on vessel gross tonnage in HSD: high speed diesel $< 5000\,\mathrm{t}$, MSD: medium speed diesel $5000-30000\,\mathrm{t}$, or SSD: slow speed diesel $> 50000\,\mathrm{t}$ categories (Lack et al., 2008, 2009; Williams et al., 2009; Diesch et al., 2013; Buffaloe et al., 2014). The other approach is to report EF for a single ship operating on specific fuel type as a function of engine load (Agrawal et al., 2008; Petzold et al., 2010, 2011; Khan et al., 2012b; Cappa et al., 2014).

#### 3.7.1 Gas pollutants

Figure 12 and Table 3 show emissions factors for $NO_x$ in this study in comparison to other studies in the literature. $EF_{NO_x}$ is expected to increase for engines operating at higher temperatures (thermal $NO_x$) (Sinha et al., 2003; Diesch et al., 2013; Cappa et al., 2014). Higher engine loads have been shown to increase $EF_{NO_x}$ (Agrawal et al., 2008; Petzold et al., 2011; Khan et al., 2012b; Cappa et al., 2014). Increasing gross tonnage has also been shown to result in higher $EF_{NO_x}$ (Williams et al., 2009; Diesch et al., 2013). $EF_{NO_x}$ in this study is in good agreement with other studies particularly for low engine loads and HSD-MSD vessel categories. However there is an increase in $EF_{NO_x}$ by a factor of 3 for plumes 2 and 3 compared to plume 1. This suggests that icebreaking during these two plumes resulted in higher engine temperatures that correspondingly increased $EF_{NO_x}$.

Figure 13 and Table 4 show emissions factors for CO in this study in comparison to other studies in the literature. Emissions factors for carbon monoxide ($EF_{CO}$) are expected to drop with increasing ship engine load (speed) (Agrawal et al., 2008; Moldanová et al., 2009; Agrawal et al., 2010; Petzold et al., 2011; Jalkanen et al., 2012; Khan et al., 2012b; Cappa et al., 2014). $EF_{CO}$ in this study is in good agreement with other studies for which the vessel speed is very slow.

#### 3.7.2 Particle pollutants

Figure 14 and Table 5 show emissions factors for rBC in this study in comparison to other studies in the literature. It is important to realize that estimates for BC measurements significantly depend on the methodology used, so caution should be used in interpreting data. For example, refractory derived SP2 measurements of BC underestimates BC emissions by a factor of about 2 relative to other techniques, likely due to methodological limiations, such as the limited range for particle detection ($60\,\mathrm{nm} < d_{p,VED} < 300\,\mathrm{nm}$) (Buffaloe et al., 2014; Cappa et al., 2014), and so where





possible, combining multiple measurement techniques for BC is desirable. With this consideration, our estimated $EF_{rBC}$ is in good agreement with other studies with low engine loading (Petzold et al., 2010, 2011). The effect of engine load on $EF_{rBC}$ has been debated in the literature. While Agrawal et al. (2008); Petzold et al. (2010, 2011); Khan et al. (2012b) find increasing emissions factors by decreasing engine loading, Cappa et al. (2014) find the opposite trend.

Figure 15 and Table 6 show emissions factors for total particle count in this study in comparison to other studies in the literature. The caveat in this comparison is the difference in lower size limit for CPC measurements. For this purpose, we have provided lower size limits for other studies. Regardless, $EF_{CPC}$ for plume 1 is higher by a factor of 4 compared to plumes 2 and 3. This suggests that higher engine loading results in higher $EF_{CPC}$. This is in agreement with studies by Petzold et al. (2010); Cappa et al. (2014) although the study by Petzold et al. (2011) has found decreasing $EF_{CPC}$ with increasing engine loading.

Figure 16 and Table 7 show emissions factors for cloud condensation nuclei in this study in comparison to other studies in the literature. The caveat in this comparison is the difference between supersaturation (SS) for CCN measurements. For this purpose SS is provided for other studies (see Table 7). $EF_{CCN}$ for the Amundsen is comparable to other studies at low engine load conditions and similar SS (Petzold et al., 2010; Cappa et al., 2014). The low $EF_{CCN}$ can be justified by the fact that there was no measurable $SO_2$ in the plumes, given the lower detection limit of our instrument, due to effective exhaust after treatment to remove this species. This suggests why CCN levels are reduced due to lack of sulphates and that the reduced CCN activity limits the ability of particles to influence clouds.

## 4 Conclusions and future work

In an effort to understand ship emissions and processing in the Arctic environment, the plume dispersion and emission factors from the Canadian Coast Guard Amundsen icebreaker were quantified near Resolute Bay, NU, Canada, during the summer 2014 NETCARE campaign. Three plumes (1, 2, and 3) were studied on consecutive days from 19 to 21 July 2014 by airborne interception using the Polar 6 aircraft, an analytical plume dispersion model, and by the FLEXPART-WRF dispersion model. The first plume measurement was carried out during the operation of Amundsen in the open water while moving at an average speed of $3.23 \pm 0.25\,\text{kts}$. The second and third plume measurements were carried out when Amundsen reached the ice edge and operated under icebreaking conditions with much lower speeds of $1.31 \pm 1.92\,\text{kts}$ and $0.09 \pm 0.30\,\text{kts}$, respectively. The engine load was low compared to cruising conditions during this campaign. The measured species included $CO_2$, $NO_x$, CO, $SO_2$, particle number concentration using a Condensation Particle Counter (CPC), refractory Black Carbon (rBC), and Cloud Condensation Nuclei (CCN). The results were compared to similar experimental studies in mid latitudes.




The calculated analytical expansion rates were $\gamma = 0.75 \pm 0.80$, $0.93 \pm 0.37$, and $1.19 \pm 0.39$ for plumes 1, 2, and 3, respectively. These are lower than observations in mid latitudes. All emission factors were in agreement with other observations at low engine loads in mid latitudes. Icebreak-

ing appeared to increase the $NO_x$ emission factor from $EF_{NO_x} = 22.3 \pm 8.0$ to $57.8 \pm 11.0$ and $65.8 \pm 4.0 \, \mathrm{g \, kg - diesel^{-1}}$ for plumes 1, 2, and 3, possibly due to high engine temperatures. The CO emission factor was $EF_{CO} = 6.4 \pm 11.7$, $6.8 \pm 2.2$ and $5.0 \pm 1.0 \, \mathrm{g \, kg - diesel^{-1}}$ for plumes 1, 2, and 3. The rBC emission factor was $EF_{rBC} = 0.20 \pm 0.04$ and $0.25 \pm 0.12 \, \mathrm{g \, kg - diesel^{-1}}$ for plumes 1 and 2. The CN emission factor was reduced while icebreaking from $EF_{CPC} = 1.96 \pm 0.41$

to $0.43 \pm 0.11$ and $0.47 \pm 0.04 \times 10^{16} \, \mathrm{kg - diesel^{-1}}$ for plumes 1, 2, and 3. The CCN emission factor was similar to observations in mid latitudes at low engine loads with $EF_{CCN} = 1.63 \pm 0.41$ to $1.06 \pm 0.32$ and $0.28 \pm 0.07 \times 10^{14} \, \mathrm{kg - diesel^{-1}}$ for plumes 1, 2, and 3.

The difference in plume expansion rate compared to mid latitude observations is attributed to unique physics of the Arctic boundary layer, which is characterized by reduced turbulent mixing

due to the thermally stable boundary layer. In addition, ship operation at partial engine load and icebreaking mode contribute to different emission factors compared to cruising conditions.

One limitation of this study was that the Amundsen plume was not intercepted at higher engine loads near cruising conditions. Future studies should measure the emission factors and plume geometrical evolution under such conditions to provide a more complete understanding of plume

chemistry and physics over the Arctic marine boundary layer.

*Acknowledgements.* The authors acknowledge a large number of people for their contributions to this work. We appreciate expert internal review of the manuscript by Pacal Bellavance and Paul Izdebski (Environment Canada - EC). We thank Tim Papakyriakou and Greg Wentworth for providing ship track and speed information. We thank Kenn Borek Air, in particular Kevin Elkes and John Bayes for their skillful piloting that facilitated

these plume observations. We are grateful to John Ford and the University of Toronto (UofT) machine shop, Jim Hodgson and Lake Central Air Services in Muskoka, Jim Watson (Scale Modelbuilders, Inc.), Julia Binder, Martin Gehrman, and Hannes Schulz (Alfred Wegener Institute - AWI - Helmholtz Center for Polar and Marine Research), Mike Harwood and Andrew Elford (EC), for their support of the integration of the instrumentation and aircraft, Yuan You (EC), for helping with data analysis, and Jeremy Wentzell (EC), for sharing an $SO_2$

mixing ratio dataset that was collected on-board of the Amundsen. We thank Mohammed Wasey for his expert support of the instrumentation during the integration and in the field. We are grateful to Stewart Cober and Carrie Taylor (EC), Bob Christensen (UofT), Kevin Riehl (Kenn Borek Air), Lukas Kandora and Jens Herrmann (AWI), Desiree Toom, Sangeeta Sharma, Dan Veber, Andrew Platt, Anne Marie Macdonald, and Maurice Watt (EC) for their support of the study. We thank the Biogeochemistry department of Max-Planck-Institut

für Chemie (MPIC) for providing the CO instrument and Dieter Scharffe for his excellent support during the preparation phase of the campaign. We thank the Nunavut Research Institute and the Nunavut Impact Review Board for licensing the study. Logistical support in Resolute Bay was provided by the Polar Continental Shelf Project (PCSP) of Natural Resources Canada for Field Project #218-14, and we are particularly grateful to Tim



McCagherty and Jodi MacGregor of the PCSP. Funding for this work was provided by the Natural Sciences and

Engineering Research Council of Canada (NSERC) under the CCAR NETCARE project, the Alfred Wegener

Institute (AWI), and Environment Canada (EC).



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





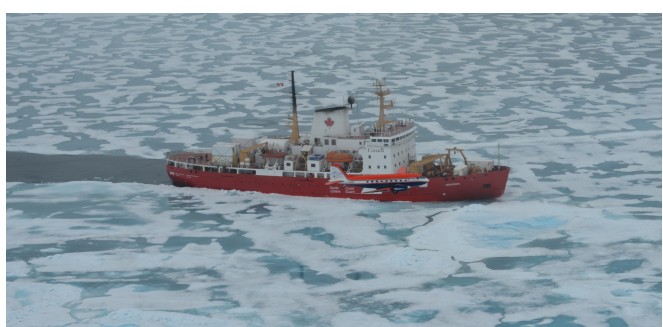

**Figure 1.** Snapshot of Polar 6 aircraft while sampling CCGS Amundsen's plume during ice breaking in Lancaster Sound (Photo credit: Maurice Levasseur).

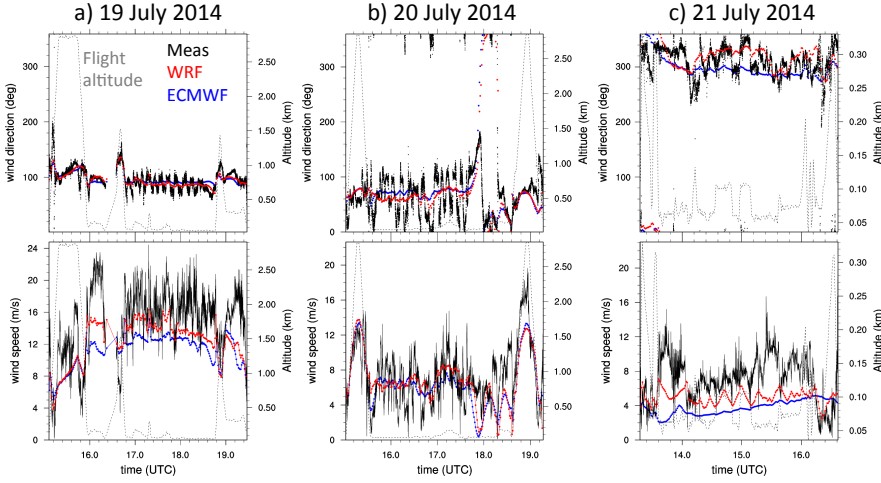

**Figure 2.** Measurements of wind direction and wind speed during the plume sampling flights (black). The modeled wind direction and wind speeds interpolated in space and time to the location of the aircraft are shown for the ECMWF analysis (blue) and WRF model (red). The flight altitude is shown in gray (dashed line).





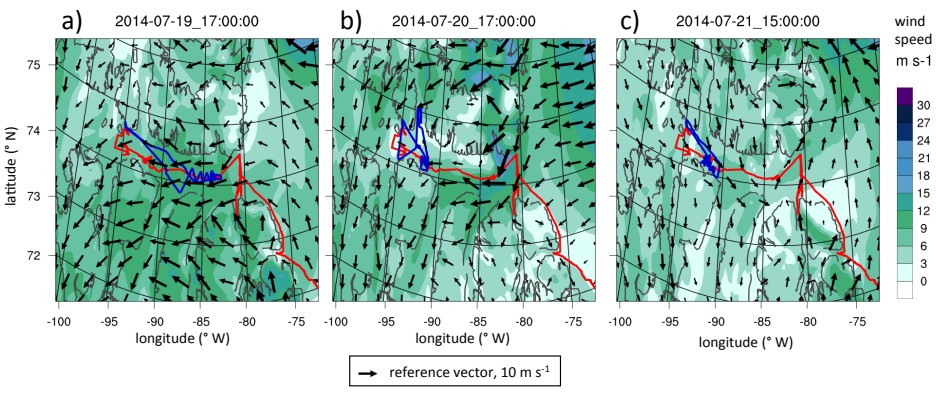

**Figure 3.** Snapshots of surface wind speed and direction predicted by WRF, with run details provided in Wentworth et al. (2015), during the Amundsen's ship emissions measurements by the Polar 6 aircraft for plumes 1 (a), 2 (b), and 3 (c). The color indicates surface wind speed and the arrow indicates both speed, with respect to the reference wind vector, and wind direction. The ship and aircraft tracks are indicated by red and blue traces, respectively.





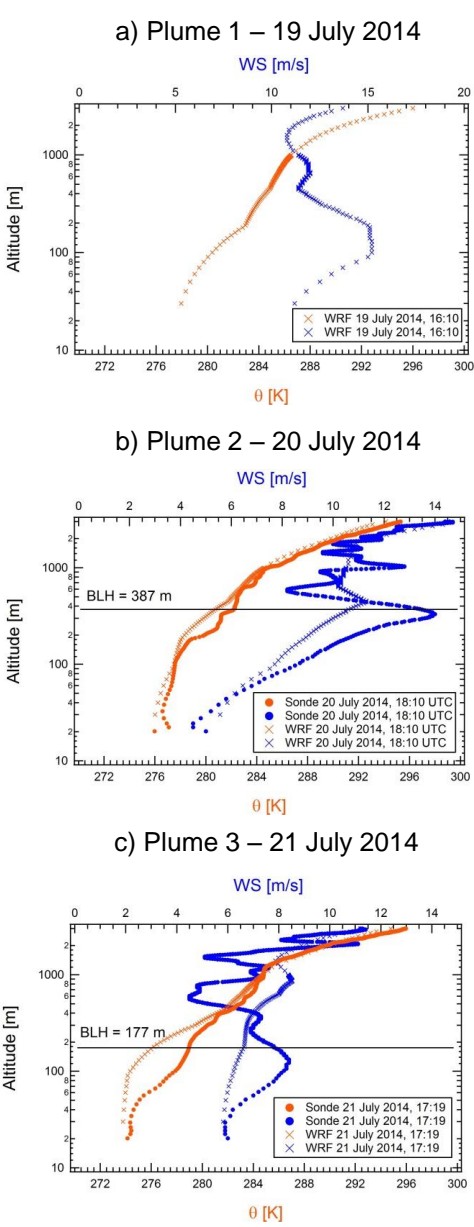

**Figure 4.** Vertical profiles of potential temperature ($\theta$) and wind speed (WS) as measured with radiosondes lunched from CCGS Amundsen and WRF (interpolated in space and time to the radiosonde launch location) during plumes 1 (a), 2 (b), and 3 (c). We note there was no radiosonde launched on 19 July 2014. The Boundary Layer Height (BLH) calculated from the measurements is shown for 20 and 21 July 2014.





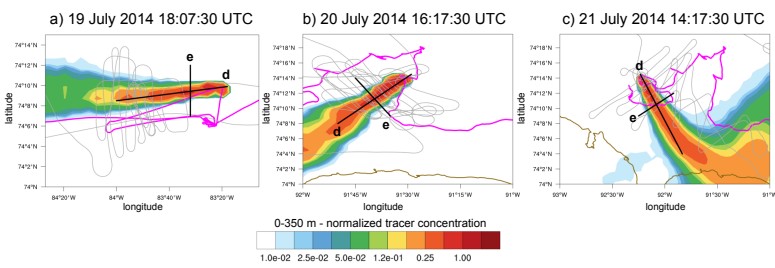

**Figure 5.** Snapshots of normalized FLEXPART-WRF predicted partial columns (0-350 m) indicating the location of the ship (initial location of emitted plume) and the predicted plume location. The flight track is shown in gray and the ship track is shown in magenta. The vertical plume structure is studied in Figure 7 along the plume (noted by black line, d) and across the plume (noted by black line, e) on each panel. Coastlines are shown in gold.

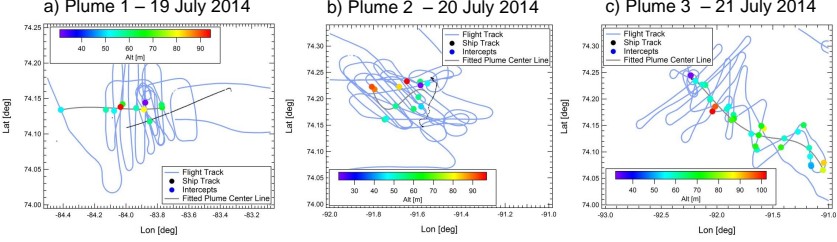

**Figure 6.** Plume location according to aircraft intercepts along the flight track identified as enhancements above background $NO_x$ mixing ratio. The plume locations identified here are in agreement with normalized FLEXPART-WRF predicted partial columns in Figure 5.





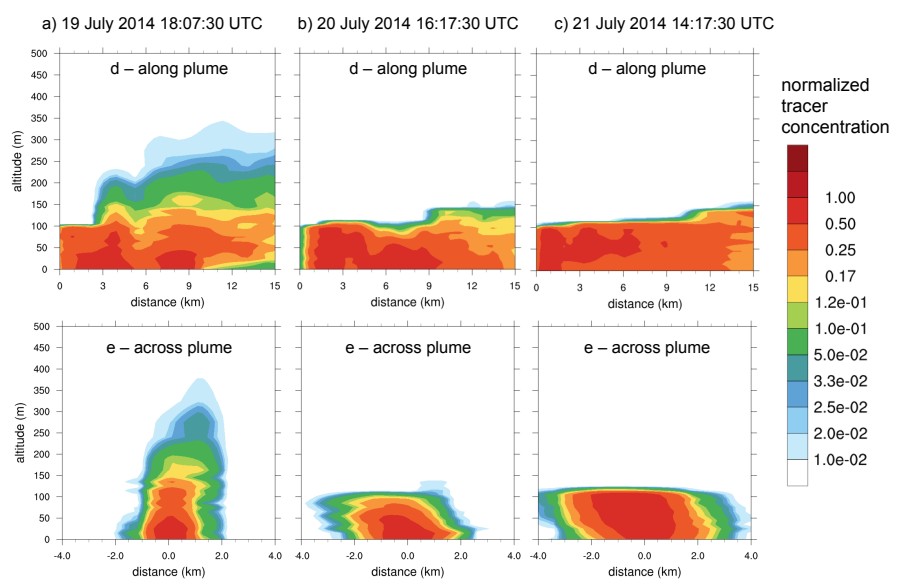

**Figure 7.** Vertical cross sections (normalized tracer concentrations) predicted by FLEXPART-WRF along plumes (panels marked d - *along plume*) and across plumes (panels marked e - *across plume*) for the same times as shown in Figure 5.



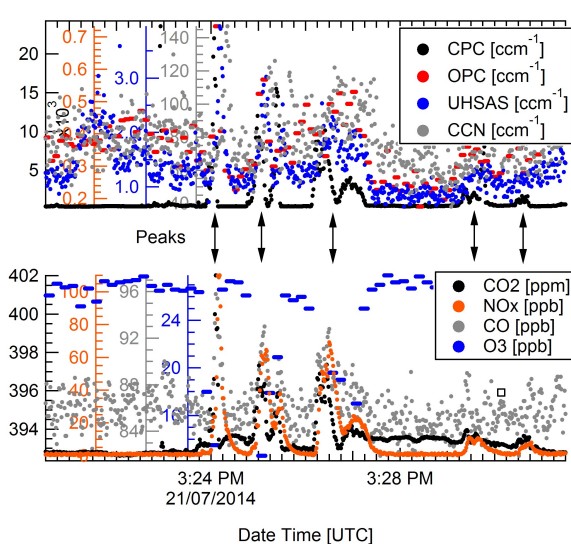

**Figure 8.** An example time series plot for identified pollution peaks in plume 3; sampling time for all instruments is 1 s except for $O_3$ (10 s) and OPC (5 s).

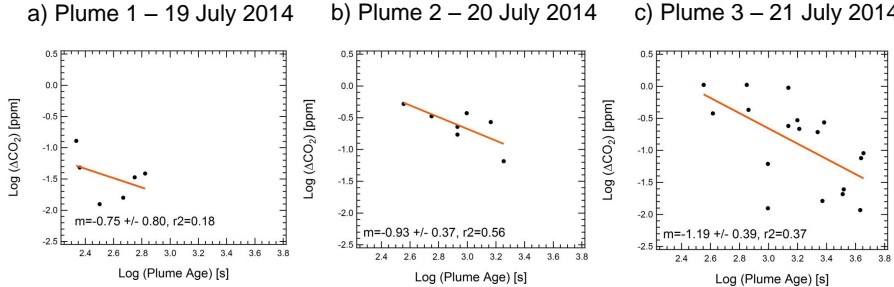

a) Plume 1 – 19 July 2014 b) Plume 2 – 20 July 2014 c) Plume 3 – 21 July 2014

**Figure 9.** Calculated plume growth or expansion rate ($\gamma = -m$) along the flight tracks using aircraft measurements for plume 1 (a), plume 2 (b), and plume 3 (b). Note: using the methodology in section 2.2.5 a plume age could be assigned to $n = 6$ data points for plume 1, $n = 7$ data points for plume 2, and $n = 18$ data points for plume 3.





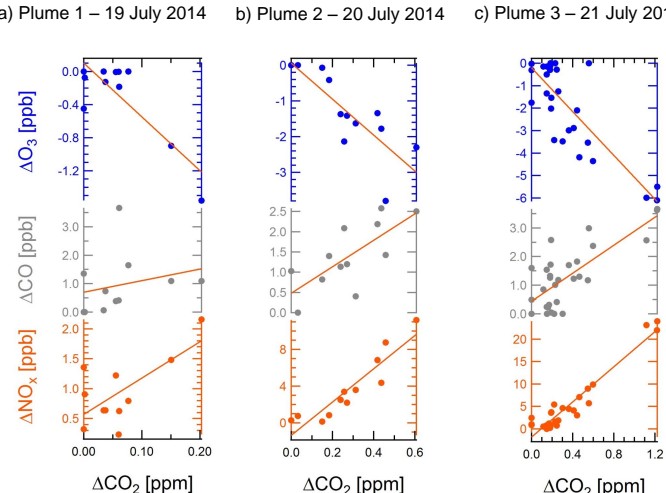

**Figure 10.** Scatter plot for excess gas pollutant mixing ratio versus excess $CO_2$ for plume 1 (a), 2 (b), and 3 (c).





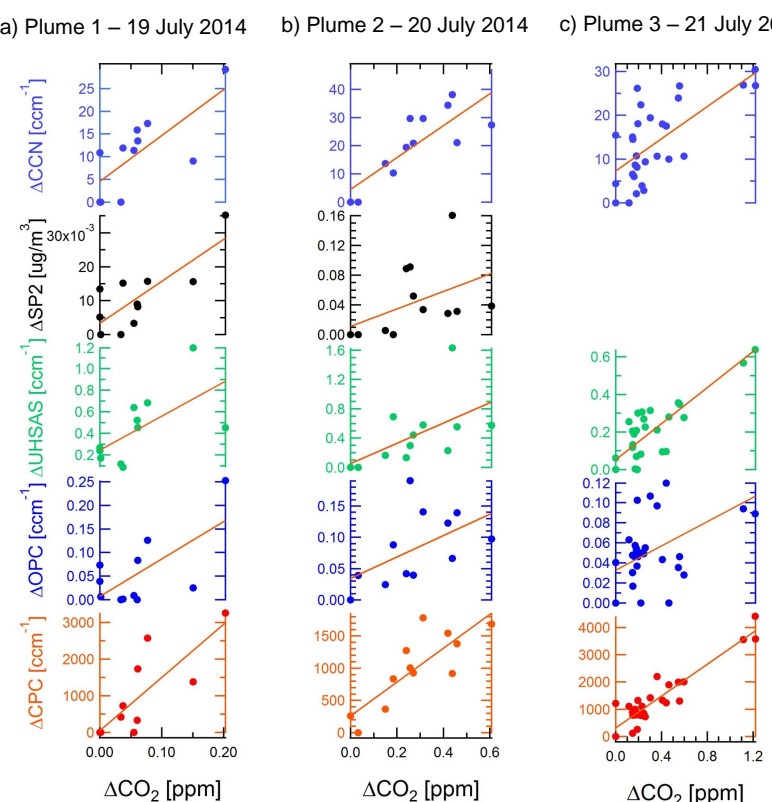

**Figure 11.** Scatter plot for excess particle concentration versus excess carbon dioxide for plume 1 (a), 2 (b), and 3 (c).





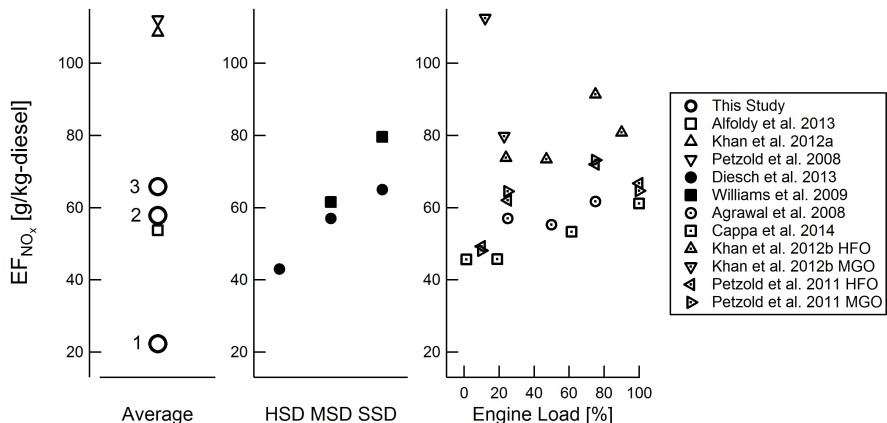

**Figure 12.** Emissions factors for $NO_x$; fuel type (HFO: heavy fuel oil with high sulfur content, and MGO: marine gas oil with low sulfur content), or vessel class based on gross metric tonnage (HSD: high speed diesel $< 5000\,t$, MSD: medium speed diesel $5000 - 30000\,t$, or SSD: slow speed diesel $> 50000\,t$); plumes 1, 2, and 3 indicated on the plot with numbers 1, 2, and 3

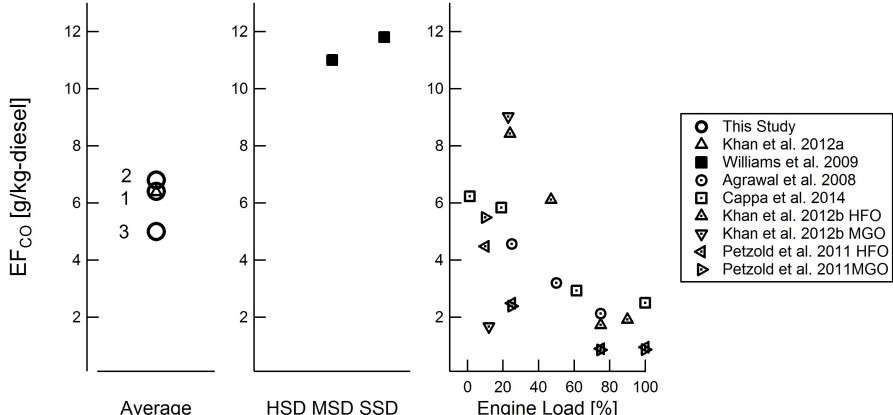

**Figure 13.** Emissions factors for CO.





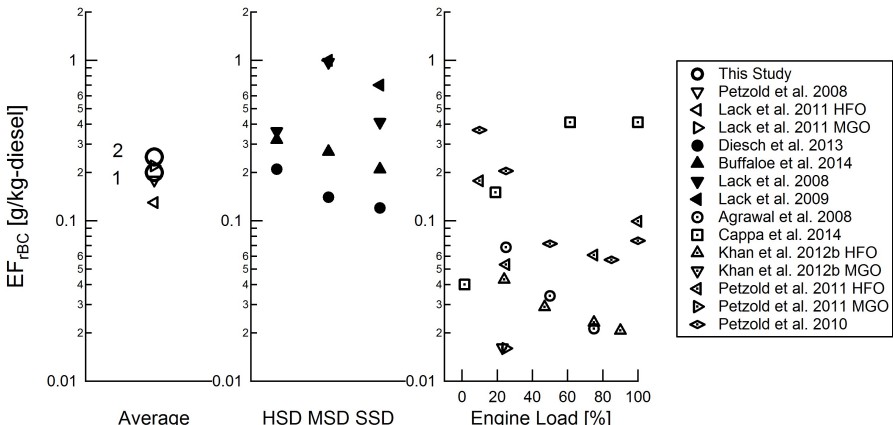

**Figure 14.** Emissions factors for black carbon.

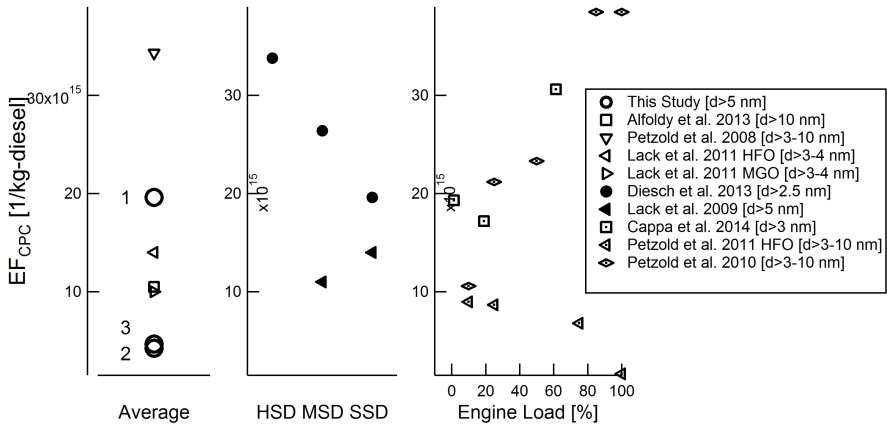

**Figure 15.** Emissions factors for for total particle count.



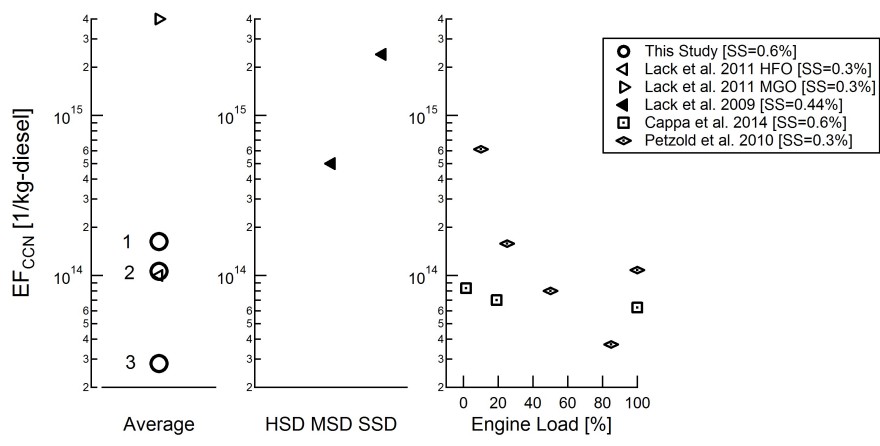

**Figure 16.** Emissions factors for cloud condensation nuclei.



**Table 1.** Linear regression analysis for excess gas pollutant mixing ratio versus excess carbon dioxide; $[\Delta X] = a + b[\Delta CO_2]$, where $X$ is any gas pollutant species; the uncertainty is computed for the regression analysis as one standard deviation.

| Species | n | $R^2$ | a [ppb] | b [ppb/ppm] |
|---|---|---|---|---|
| | | Plume 1 | | |
| $\Delta$ NO$_x$ | 11 | 0.47 | 0.57±0.19 | 6.1±2.2 |
| $\Delta$ CO | 11 | 0.06 | 0.69±0.47 | 4.1±5.4 |
| $\Delta$ O$_3$ | 11 | 0.69 | 0.10±0.13 | -6.5±1.5 |
| | | Plume 2 | | |
| $\Delta$ NO$_x$ | 12 | 0.85 | -1.4±0.81 | 18±2.5 |
| $\Delta$ CO | 12 | 0.52 | 0.48±0.33 | 3.3±0.99 |
| $\Delta$ O$_3$ | 12 | 0.64 | 0.060±0.39 | -5.0±1.2 |
| | | Plume 3 | | |
| $\Delta$ NO$_x$ | 29 | 0.92 | -1.8±0.54 | 19±1.1 |
| $\Delta$ CO | 29 | 0.57 | 0.44±0.19 | 2.4±0.41 |
| $\Delta$ O$_3$ | 29 | 0.67 | -0.20±0.31 | -4.9±0.65 |



**Table 2.** Linear regression analysis for excess particle concentration versus excess carbon dioxide; $[\Delta X] = a + b[\Delta CO_2]$, where $X$ is any particle concentration; the uncertainty is computed for the regression analysis as one standard deviation.

| Species | n | $R^2$ | a [ccm$^{-1}$] or [μg m$^{-3}$] | b [ccm$^{-1}$/ppm] or [μg m$^{-3}$/ppm] |
|---|---|---|---|---|
| | | | Plume 1 | |
| $\Delta$ CPC [ccm$^{-1}$] | 11 | 0.65 | 51±300 | 15000±3500 |
| $\Delta$ OPC [ccm$^{-1}$] | 11 | 0.43 | 0.0070±0.027 | 0.79±0.31 |
| $\Delta$ UHSAS [ccm$^{-1}$] | 11 | 0.39 | 0.24±0.12 | 3.2±1.3 |
| $\Delta$ SP2 [μg m$^{-3}$] | 11 | 0.62 | 0.0033±0.0028 | 0.12±0.03 |
| $\Delta$ CCN [ccm$^{-1}$] | 11 | 0.54 | 4.6±2.7 | 100±30 |
| | | | Plume 2 | |
| $\Delta$ CPC [ccm$^{-1}$] | 12 | 0.68 | 260±190 | 2600±600 |
| $\Delta$ OPC [ccm$^{-1}$] | 12 | 0.29 | 0.034±0.028 | 0.17±0.085 |
| $\Delta$ UHSAS [ccm$^{-1}$] | 12 | 0.31 | 0.054±0.21 | 1.4±0.65 |
| $\Delta$ SP2 [μg m$^{-3}$] | 12 | 0.19 | 0.011±0.025 | 0.12±0.077 |
| $\Delta$ CCN [ccm$^{-1}$] | 12 | 0.66 | 4.5±4.3 | 56±13 |
| | | | Plume 3 | |
| $\Delta$ CPC [ccm$^{-1}$] | 29 | 0.86 | 310±100 | 2900±200 |
| $\Delta$ OPC [ccm$^{-1}$] | 29 | 0.30 | 0.033±0.0086 | 0.061±0.018 |
| $\Delta$ UHSAS [ccm$^{-1}$] | 29 | 0.74 | 0.052±0.026 | 0.48±0.054 |
| $\Delta$ CCN [ccm$^{-1}$] | 29 | 0.46 | 7.3±1.8 | 18±3.9 |





**Table 3.** Emissions factors for $NO_x$; numbers in brackets indicate engine load (%), fuel type (HFO: heavy fuel oil with high sulfur content, and MGO: marine gas oil with low sulfur content), or vessel class based on gross metric tonnage (HSD: high speed diesel $< 5000\,\mathrm{t}$, MSD: medium speed diesel $5000-30000\,\mathrm{t}$, or SSD: slow speed diesel $> 50000\,\mathrm{t}$).

| Study | $EF_{NO_x}$ [g kg $-$ diesel$^{-1}$] (NO$_2$ equivalent) |
|---|---|
| This Study | 22.3±8.0 (Plume 1) 57.8±11.0 (Plume 2) 65.8±4.0 (Plume 3) |
| Agrawal et al. (2008) | 57.0 (25 %), 55.3 (50 %), 61.7 (75 %) |
| Alföldy et al. (2013) | 53.7 |
| Cappa et al. (2014) | 45.6±8.2 (1.4 %), 45.7±8.2 (19.0 %), 53.3±9.6 (61.4 %), 61.1±11.0 (100 %) |
| Diesch et al. (2013) | 43±29 (HSD), 57±28 (MSD), 65±23 (SSD) |
| Khan et al. (2012a) | 108.5 |
| Khan et al. (2012b) | 112.4 (12 %, MGO), 79.75 (23 %, MGO) |
| Khan et al. (2012b) | 73.8 (24 %, HFO), 73.4 (47 %, HFO), 91.4 (75 %, HFO), 80.8 (90 %, HFO) |
| Petzold et al. (2011) | 64.7 (100 %, MGO), 73.1 (75 %, MGO), 64.5 (25 %, MGO), 48.1 (10 %, MGO) |
| Petzold et al. (2011) | 66.7 (100 %, HFO), 72.0 (75 %, HFO), 62.0 (25 %, HFO), 49.2 (10 %, HFO) |
| Petzold et al. (2008) | 112 |
| Williams et al. (2009) | 61.5±22.9 (MSD), 79.6±27.4 (SSD) |

**Table 4.** Emissions factors for CO.

| Study | $EF_{CO}$ [g kg $-$ diesel$^{-1}$] |
|---|---|
| This Study | 6.4±11.7 (Plume 1) 6.8±2.2 (Plume 2) 5.0±1.0 (Plume 3) |
| Agrawal et al. (2008) | 4.6 (25 %), 3.2 (50 %), 2.1 (75 %) |
| Cappa et al. (2014) | 6.23±1.2 (1.4 %), 5.83±0.9 (19.0 %), 2.92±0.58 (61.4 %), 2.50±0.5 (100 %) |
| Khan et al. (2012a) | 6.38 |
| Khan et al. (2012b) | 1.7 (12 %, MGO), 9.0 (23 %, MGO) |
| Khan et al. (2012b) | 8.4 (24 %, HFO), 6.11 (47 %, HFO), 1.7 (75 %, HFO), 1.9 (90 %, HFO) |
| Petzold et al. (2011) | 0.86 (100 %, MGO), 0.85 (75 %, MGO), 2.39 (25 %, MGO), 5.49 (10 %, MGO) |
| Petzold et al. (2011) | 0.95 (100 %, HFO), 0.89 (75 %, HFO), 2.49 (25 %, HFO), 4.47 (10 %, HFO) |
| Williams et al. (2009) | 11.0±14.2 (MSD), 11.8±11.7 (SSD) |



**Table 5.** Emissions factors for black carbon; [a] elemental carbon, filter measurement based on a thermal/optical carbon aerosol analyzer according to NIOSH 5040; [b] black carbon measurement based on weighted average using SP2, SP-AMS, PAS, and PSAP; [c] black carbon measurement based on weighted average using SP2, SP-AMS, PAS-G, PAS-B, and PSAP; [d] black carbon measurement based on Multiple Angle Absorption Photometer (MAAP); [e] light absorbing carbon measurement based on photoacoustic techniques; [f] black carbon measurement based on PAS; [g] elemental carbon, filter measurement based on a multi-step combustion method according to VDI guideline 2465-2.

| Study | $EF_{rBC}$ [g kg $-$ diesel$^{-1}$] |
|---|---|
| This Study | 0.20±0.04 (Plume 1) 0.25±0.12 (Plume 2) |
| Agrawal et al. (2008)[a] | 0.068 (25 %), 0.034 (50 %), 0.021 (75 %) |
| Buffaloe et al. (2014)[b] | 0.32±0.26 (HSD), 0.27±0.12 (MSD), 0.21±0.16 (SSD) |
| Cappa et al. (2014)[c] | 0.04 (1.4 %), 0.15 (19.0 %), 0.41 (61.4 %), 0.41 (100 %) |
| Diesch et al. (2013)[d] | 0.21±0.23 (HSD), 0.14±0.16 (MSD), 0.12±0.08 (SSD) |
| Khan et al. (2012b)[a] | 0.010 (12 %, MGO), 0.016 (23 %, MGO) |
| Khan et al. (2012b)[a] | 0.043 (24 %, HFO), 0.029 (47 %, HFO), 0.023 (75 %, HFO), 0.021 (90 %, HFO) |
| Lack et al. (2009)[e] | 1.0±0.7 (MSD), 0.7±0.8 (SSD) |
| Lack et al. (2011)[f] | 0.22±0.09 (MGO), 0.13±0.05 (HFO) |
| Lack et al. (2008)[e] | 0.36±0.23 (HSD), 0.97±0.66 (MSD), 0.41±0.27 (SSD) |
| Petzold et al. (2011)[d] | 0.005 (100 %, MGO), 0.006 (75 %, MGO), 0.016 (25 %, MGO), 0.007 (10 %, MGO) |
| Petzold et al. (2011)[d] | 0.099 (100 %, HFO), 0.061 (75 %, HFO), 0.053 (25 %, HFO), 0.178 (10 %, HFO) |
| Petzold et al. (2010)[d] | 0.075 (100 %), 0.057 (85 %), 0.072 (50 %), 0.204 (25 %), 0.367 (10 %) |
| Petzold et al. (2008)[g] | 0.179±0.018 |

**Table 6.** Emissions factors for total particle count $\times 10^{16}$.

| Study | $EF_{CPC}$ [kg $-$ diesel$^{-1}$] |
|---|---|
| This Study [$d > 5$ nm] | 1.96±0.41 (Plume 1) 0.43±0.11 (Plume 2) 0.47±0.04 (Plume 3) |
| Alföldy et al. (2013) [$d > 10$ nm] | 1.05±0.10 |
| Cappa et al. (2014) [$d > 3$ nm] | 1.93 (1.4 %), 1.72 (19.0 %), 3.06 (61.4 %), 2.23 (100 %) |
| Diesch et al. (2013) [$d > 2.5$ nm] | 3.38±3.1 (HSD), 2.64±0.15 (MSD), 1.96±0.70 (SSD) |
| Lack et al. (2009) [$d > 5$ nm] | 1.1±0.8 (MSD), 1.4±1.0 (SSD) |
| Lack et al. (2011) [$d > 3 - 4$ nm] | 1.0±0.2 (MGO), 1.4±0.2 (HFO) |
| Petzold et al. (2011) [$d > 3 - 10$ nm] | 0.17 (100 %, HFO), 0.68 (75 %, HFO), 0.87 (25 %, HFO), 0.90 (10 %, HFO) |
| Petzold et al. (2010) [$d > 5$ nm] | 3.85±0.30 (100 %), 3.85±0.17 (85 %), 2.33±0.18 (50 %), 2.12±0.09 (25 %), 1.06±0.10 (10 %) |
| Petzold et al. (2008) [$d > 3 - 10$ nm] | 3.43±1.26 |





**Table 7.** Emissions factors for cloud condensation nuclei $\times 10^{14}$.

| Study | $EF_{CCN}$ [kg $-$ diesel$^{-1}$] |
|---|---|
| This Study [SS=0.6 %] | 1.63±0.41 (Plume 1) 1.06±0.32 (Plume 2) 0.28±0.07 (Plume 3) |
| Cappa et al. (2014) [SS=0.6 %] | 0.83 (1.4 %), 0.7 (19.0 %), 0.63 (100 %) |
| Lack et al. (2009) [SS=0.44 %] | 5.0±3.0 (MSD), 24±20 (SSD) |
| Lack et al. (2011) [SS=0.3 %] | 40±4 (MGO), 1.0±0.1 (HFO) |
| Petzold et al. (2010) [SS=0.3 %] | 1.08 (100 %), 0.37 (85 %), 0.80 (50 %), 1.58 (25 %), 6.15 (10 %) |