# Peer review of "Ship emissions measurement in the Arctic by plume intercepts of the Canadian Coast Guard icebreaker *Amundsen* from the *Polar 6* aircraft platform"

_Atmospheric Chemistry and Physics, 2015_

## Referee Comment (RC1) · Anonymous Referee #1 · 30 Mar 2016

General comments

This manuscript presents airborne observations of emission plumes from an icebreaking ship operating in Arctic waters, and allows the authors an opportunity to contribute to the existing body of literature on ship emissions. Specifically, the authors present calculations of plume expansion parameters in the Arctic boundary layer, and emission factors for an icebreaker under different operation conditions (open water vs. icebreaking operations). Both the emission factors and the plume parameters are important for the modeling of ship exhaust impacts in the Arctic. I found the scientific

significance and scientific quality of the manuscript to be good, and appropriate for publication. However, I currently find the presentation quality of the manuscript to be lacking, and a significant revision will be required before this manuscript should be accepted. Many of the figures will need to be improved for publication. Figures 2, 5, 6 use traces and axis labels that are too thin or light in color to be seen at the print size of the figure. Figure 8 has multiple vertical axis covered by scatter points and therefore cannot be read. More generally, I found that viewing the manuscript pdf at 300% magnification was necessary to begin to interpret the figures.

Specific Comments

Section 2.1

Can the authors categorize the Amundson by engine category (HSD/MSD/LSD) and fuel type (HFO/MGO)? This would be useful for the comparison to literature later in the text.

Section 2.2.2

In light of the small changes in mixing ratios of $CO_2$ and CO reported, perhaps a discussion of precision of these measurements is important to assess the data quality used for the linear fits. It seems to me that stating only total uncertainty may undersell the quality of the data presented here.

Section 2.2.5

The discussion of the estimation of plume age is difficult to follow, in part because Figure 6 is very hard to interpret. The legend indicates large black circles for the ship track, which I don't see. Also the flight track and fitted plume center line are indistinguishable to me (I'm color-blind). I don't see where the intercept height is used anywhere in the analysis, and so can be removed from the figure. A modest proposal would be to simply color the flight track by $NO_x$ mixing ratio so the horizontal extent of the plume intercepts can be seen.

[Figure]

Parsing the text, I don't understand the description of equation 3. The plume center line is described as a high-order polynomial fit of the observed plume intercepts. The plume center line origin (l = 0) is not defined. Is it the ship track? Please add an additional sentence or two here to make this a bit more clear.

Section 3.1

Line 274, "flight tracks shown in Figure 2." Figure 2 shows time traces of wind speed and wind direction, not flight tracks. The measurements of wind speed and direction seem to have a bias with aircraft flight direction – at least this is how I'm interpreting the oscillations shown, especially in panels a) and b). Was this corrected for the plume age calculations? Can vertical lines or another marker be added to show when the plumes were intercepted?

Section 3.3

FLEXPART uses a 100m vertical and horizontal emission source, and the modeled plume structures are presented in Figure 7. For panels b) and c), the plume height shows very little increase across 15 km but considerable growth of plume width. The measured and modeled boundary layer heights for those days are higher than the plume heights, implying very slow vertical mixing. How does this compare with von Glasow [2003], who estimate $\beta \approx 0.6$ for the temperate MBL? FLEXPART seems to show $\alpha \gg \beta$ for panels b) and c), but not a). Can the authors discuss this, especially within the context of their observations? If $\alpha \gg \beta$ is true for b) and c) but not a), would this have an impact upon the calculation of $\gamma$ in Section 3.5?

In light of the large uncertainty of the fitted line for the first plume intercept, can the authors discuss the uncertainty of the plume age calculation relative to the uncertainty of $CO_2$ enhancement?

Section 3.5

The interpretation of plume expansion rates in the boundary layer (BL) hinges upon

whether the plume has mixed vertically to the top of the BL. I may have missed it, but I can find no observations of the extent of plume vertical mixing, but rather a reliance upon FLEXPART. It strikes me as circular logic to use modeled vertical mixing to interpret the observed plumes and thereby calculate mixing parameters that will be used by models. Could the authors use a mass-balance approach (Ryerson et al., 1998; White et al., 1976) to the observed mixing ratios during plume intercepts to evaluate the extent of vertical mixing? Can observed plume widths be compared to the modeled widths, to at least confirm that horizontal mixing is correct in FLEXPART? Can the authors make any calculation of $\alpha$ rather than just $\gamma$?

Section 3.6

Figure 8 shows changes in CO2 mixing ratio of approximately 3 – 7 ppm CO2 for the three plume intercepts shown, but the x-axis of the scatterplots for plume 3 [Figure 10, panel c) and Figure 11, panel c)] have a dynamic range of 1.2 ppm. The data shown in Figure 8 also includes many more data point than is shown in Figures 10 and 11, panel c). Am I confusing things here? Are the scatterplots (Fig 10 and 11) showing something other than the 1s data shown in Figure 8?

The fits of O3 vs CO2 have no relevance to either calculations of emission factors or plume dispersion, and should be removed.

The authors should explain how the uncertainties were calculated for the slopes of the fitted lines presented in Table 1. It appears that the fits are linear regressions, while bivariate least squares fits would be appropriate – both x and y variables have observational errors. Also, the individual uncertainties of the data points is known and can be used to weight the data. Finally, the stated uncertainty for CO2 is 0.3 ppm, while the range of enhanced CO2 mixing ratios for plume 1 is only 0.2 ppm, i.e. not significant. The data appears to show meaningful correlations, so I suspect the instrumental uncertainties are overly conservative.

Section 3.7

It appears that Tables 3-7 are redundant with Figures 12-16. The authors should use only one or the other to present this data. Figures 12-16 would also be well-served to include the uncertainty range for the authors' data points, if not for all data.

The NOX EF reported for plume 1 (open water) appears anomalously low compared to the existing literature. Can the authors discuss whether this is due to their ability to interpret the measured data, or if this is reasonable for the ship under the operating conditions at the time?

Technical Corrections

Figure 15 caption has a typo ("for for").

Ryerson, T. B., Buhr, M. P., Frost, G. J., Goldan, P. D., Holloway, J. S., Hubler, G., Jobson, B. T., Kuster, W. C., McKeen, S. A., Parrish, D. D., Roberts, J. M., Sueper, D. T., Trainer, M., Williams, J., and Fehsenfeld, F. C.: Emissions lifetimes and ozone formation in power plant plumes, Journal of Geophysical Research-Atmospheres, 103, 22569-22583, 1998.

White, W., Anderson, J., Blumenthal, D., Husar, R., Gillani, N., Husar, J., and Wilson, W.: Formation and transport of secondary air pollutants: ozone and aerosols in the St. Louis urban plume, Science, 194, 187-189, 10.1126/science.959846, 1976.

---

## Referee Comment (RC2) · Anonymous Referee #2 · 16 Apr 2016

GENERAL REMARKS

The manuscript presents results from an in-depth study on emissions from a single icebreaker vessel operating in Arctic waters. Although there is already extensive information available on the gaseous and particulate emissions from marine Diesel engines, this study adds significant data on ship emissions in the Arctic. The study is well designed and carefully conducted. It covers both the experimental determination of emission factors and the investigation of ship emissions transport in the marine boundary layer under the specific conditions of Arctic waters.

[Figure]

The manuscript is in the scope of the journal and the presentation of the material is clear and well structured. Few minor revisions should be considered before publication in ACP. Technically, most of the figures require careful revision because they are hard to read in the current version.

SPECIFIC COMMENTS

1| The only major comment concerns the determination of emission factors from the plume encounters. In the present form, the authors report average emission factors for each investigated species. However, this way of presenting the data neglects potential effects of transformation of the species during plume expansion and atmospheric transformation. Combining the information shown in Figures 9, 10, and 11, it becomes evident that plume encounters were measured at plume ages between 100 and > 1000 s. This span of plume ages should allow the investigation of emission factors at various plume ages, i.e., the determination of effective emission factors. It is recommended to determine emission factors separately for all plume encounters and plot the data as a function of plume age. Doing this would permit investigating whether there was chemical or physical transformation of the investigated species during plume ageing; see for example the results on particle ageing shown in Petzold et al. (2008) and Tian et al. (2014).

2| In the section on applied particle instruments, the information on lower detection sizes is missing. Reporting aerosol data as "CPC", "OPC" or "UHSAS" is misleading. Instead they should be reported as, e.g., $N_{(OPC, D_p 250 nm)}$ etc. Then the reader recognizes directly the aerosol mode covered by the respective instrument. Furthermore, minimum detection diameters of the instruments should be added to Section 2.2.3.

3| Careful revision of all figures is strongly recommended. They should be made much simpler and need to be enlarged. Figures 12 to 16 are not really necessary since the information is already contained in the extensive tables.

[Figure]

4| The presentation of emission factors in Tables 3 – 6 should focus on the comparison of reported values with data for similar engine types. Currently, the tables list all values available in literature without giving information whether or not the investigated engines are comparable to the engines operated aboard the Amundsen. Giving more weight to those engines of similar types would increase the readability of Tables 3 – 6 significantly.

MINOR REMARKS

Page 1, line 7: suggested rephrasing: "Canadian Coast Guard icebreaker Amundsen".

Page 6, line 169: change to "(OPC GRIMM Model 1.129).

REFERENCES

Petzold, A., Hasselbach, J., Lauer, P., Baumann, R., Franke, K., Gurk, C., Schlager, H., and Weingartner, E.: Experimental studies on particle emissions from cruising ship, their characteristic properties, transformation and atmospheric lifetime in the marine boundary layer, Atmos. Chem. Phys., 8, 2387–2403, 2008.

Tian, J., Riemer, N., West, M., Pfaffenberger, L., Schlager, H., and Petzold, A.: Modeling the evolution of aerosol particles in a ship plume using PartMC-MOSAIC, 14, 5327-5347, doi: 10.5194/acp-14-5327-2014, 2014.

---

## Author Comment (AC1) · 2 Jun 2016

Responses are uploaded in PDF format as a supplement.

Please also note the supplement to this comment:
http://www.atmos-chem-phys-discuss.net/acp-2015-1032/acp-2015-1032-AC1-supplement.pdf

---

## Author Comment (AC2) · 5 Jun 2016

In the previous author responses, the precision of $CO_2$ measurements in section 2.2.2 was inconveniently left out. The complete paragraph is corrected in the manuscript and provided below:

"During the ship emission measurements the $CO_2$ (CO) data achieved a precision (1 sigma, 1 Hz) of 0.02 ppmv (2.3 ppbv). The stability of the instrument was calculated to 0.62 ppmv (4.7 ppbv), respectively, before applying the post flight data correction. Note that stability was based on the mean drift between two subsequent calibrations which

were performed during the flights. The stability was mainly affected by temperature variations. These instrumental drifts were corrected after the flights assuming a linear drift. Hence, the total uncertainty relative to the working standard of 0.62 ppmv (5.23 ppbv) could be regarded as an upper limit. As can be seen in figure 8, the instrument precision of the CO2 and CO measurements allowed for identifying the ship emission plumes and further calculations of emission factors."

---

## Author Response (AR1)

**Ship emissions measurement in the Arctic from plume intercepts of the Canadian Coast Guard icebreaker *Amundsen* from the *Polar 6* aircraft platform**

By A. A. Aliabadi* et al.
*Massachusetts Institute of Technology, Department of Architecture,
Cambridge, USA

Response to reviewers
Manuscript ID: acp-2015-1032

June 2, 2016

Dear Dr. Harald Saathoff

We thank you and the reviewers for your time and the constructive comments toward the improvement of the manuscript acp-2015-1032. We have addressed all issues in the responses below and will upload all necessary files to the portal, including a highlighted version of the manuscript in PDF format to communicate changes effectively and easily. Below you can find point-by-point responses to the reviewer comments. The comments are shown in plain text, the responses are shown in *italics*, and the modified text in the manuscript is shown in **bold face**.

We have made our best effort to accommodate all suggestions. Briefly, we have improved all of the figures to have less clutter, used larger marker sizes and line widths, used larger fonts for axis labels and legends, and used vivid colors to encode information effectively. We have used a new multivariate least square data fitting method that takes into account variability in both $x$ and $y$ data in order to fit a line through the data-points toward the estimation of the emission factors. This method is called orthogonal distance regression (ODR) and results in more realistic emission factors. The notation for particle number and mass concentrations and emission factors have been changed. Atmospheric boundary layer heights have been estimated by both radiosonde and WRF model data to support the fact that the ship plumes did not reach the top of the boundary layer.

We do respect the comments by reviewers that there is some redundancy between Figs. 12-16 and Tables 3-6. However, in our opinion these figures and tables serve different purposes and are complementary. The tables communicate all numerical data and uncertainties in our measurements and the literature for future comparisons in other studies, but they are difficult to read. On the

other hand, the figures are meant to provide visual aid to help the reader compare our results against other previous measurements based on ship tonnage classification, engine loading, and fuel type. Therefore, in the interest of a more comprehensive and readable paper, we suggest to keep both figures and tables. Finally, it was suggested that we resolve emission factors as a function of plume age and that we measure the vertical versus horizontal growth rate of the plume. However, in our responses we reason that, although we attempted, these detailed studies were not feasible because we only had a limited number of intercepts with short time durations. In order to perform the detailed studies we must have sampled the plume more rigorously at multiple distances downwind of the plume and numerous locations both horizontally and vertically, which was not possible due to the logistics of the field campaign. Therefore, we suffice to only calculate average emission factors and combined horizontal-vertical plume growth rates.

In conclusion, we thank you for your time and trust your judgment in the final evaluation of this revised manuscript.

Best Regards,

Amir A. Aliabadi, on behalf of all co-authors

**Reviewer 1**

**General Comments**

This manuscript presents airborne observations of emission plumes from an icebreaking ship operating in Arctic waters, and allows the authors an opportunity to contribute to the existing body of literature on ship emissions. Specifically, the authors present calculations of plume expansion parameters in the Arctic boundary layer, and emission factors for an icebreaker under different operation conditions (open water vs. icebreaking operations). Both the emission factors and the plume parameters are important for the modeling of ship exhaust impacts in the Arctic. I found the scientific significance and scientific quality of the manuscript to be good, and appropriate for publication. However, I currently find the presentation quality of the manuscript to be lacking, and a significant revision will be required before this manuscript should be accepted. Many of the figures will need to be improved for publication. Figures 2, 5, 6 use traces and axis labels that are too thin or light in color to be seen at the print size of the figure. Figure 8 has multiple vertical axis covered by scatter points and therefore cannot be read. More generally, I found that viewing the manuscript PDF at 300% magnification was necessary to begin to interpret the figures.

*Response: thank you. We have improved the quality of all figures by using larger markers, thicker lines, and more vivid colors, increasing font size for axes labels and legends, and reducing clutter.*

**Specific Comments**

**Section 2.1**

Can the authors categorize the Amundsen by engine category (HSD/MSD/LSD) and fuel type (HFO/MGO)? This would be useful for the comparison to literature later in the text.

*Response: we have added the following statement at the end of section 2.1:*

**According to the conventions for ship classifications, which are to be discussed in section 3.7, the Amundsen is a medium speed diesel (MSD) ship that burns marine gas oil (MGO) with a low sulfur content.**

**Section 2.2.2**

In light of the small changes in mixing ratios of $CO_2$ and CO reported, perhaps a discussion of precision of these measurements is important to assess the data quality used for the linear fits. It seems to me that stating only total uncertainty may undersell the quality of the data presented here.

*Response: thank you. We added the precision and stability information in section 2.2.2. The following text was added at the end of the section with the $CO_2$ and CO measurements:*

**During the ship emission measurements the $CO_2$ (CO) data achieved a precision (1 sigma, 1 Hz) of 2.3 ppbv. The stability of the instrument was calculated to 4.7 ppbv, respectively, before applying the post flight data correction. Note that stability was based on the mean drift between two subsequent calibrations which were performed during the flights. The stability was mainly affected by temperature variations. These instrumental drifts were corrected after the flights assuming a linear drift. Hence, the total uncertainty relative to the working standard of 5.23 ppbv could be regarded as an upper limit. As can be seen in figure 8, the instrument precision of the $CO_2$ and CO measurements allowed for identifying the ship emission plumes and further calculations of emission factors.**

*It is important to note that the stability calculated as the mean difference of the mixing ratio between two subsequent calibrations is mainly dominated by temperature variations. During the flights the instruments slowly and steadily warmed up or cooled down to a new equilibrium temperature leading to the drift. Regular calibrations of the instrument every 15 to 30 min, which were performed at a time interval shorter than the temperature variations, allow for correcting this drift and thus reducing the accuracy error (not to be confused with precision error). Nevertheless, the precision being most important for the detection of enhancements due to ship emissions is unaffected from these temperature effects.*

**Section 2.2.5**

The discussion of the estimation of plume age is difficult to follow, in part because Fig. 6 is very hard to interpret. The legend indicates large black circles for the ship track, which I don't see. Also the flight track and fitted plume center line are indistinguishable to me (I'm color-blind). I don't see where the intercept height is used anywhere in the analysis, and so can be removed from the figure. A modest proposal would be to simply color the flight track by $NO_x$ mixing ratio so the horizontal extent of the plume intercepts can be seen.

*Response: thank you. Regarding figure 6 we have improved the quality of this figure significantly. We have now used a diverse and vivid color composition for different legends in this figure. The ship track is shown by large blue markers that are packed and appear as a continuous path. The flight track is shown by slightly smaller green markers that are also packed and appear as a continuous path. The center line of the plume is shown by large black markers that are also packed and appear as a continuous path. The present color choices and the increased marker sizes now make distinguishing the ship track, flight track, and plume center line easier. As suggested, we have removed altitude information from the figure because it was not used anywhere in the study, except for the observation that the plumes did not reach the top of the boundary layer. We originally tried color coding the flight track by $NO_x$ mixing ratio, however the large variations in peak amplitudes and very short intercept times resulted in indistinguishable intercept locations. Instead, we picked the latitude and longitude associated with the center of each $NO_x$ peak and used a new legend (large red circles) in the plot that now appears as a distinct identifier for plume intercept events.*

Parsing the text, I don't understand the description of equation 3. The plume center line is described as a high-order polynomial fit of the observed plume intercepts. The plume center line origin ($l = 0$) is not defined. Is it the ship track? Please add an additional sentence or two here to make this a bit more clear.

*Response: we appreciate this point. We have explained more carefully and with more detail the methodology used for estimating plume age and identifying the plume origin on the center line. The text in section 2.2.5 has been modified to:*

**The wind measurements on board of the aircraft along the flight track and closest to each point on the center line were then used to estimate wind velocity along the plume center line. The plume age was estimated at each intercept by calculating the time it took for a parcel of air from the plume origin on the center line ($l = 0$) to travel a distance of $l = L$ along the centerline and reach the nearest location to the intercept of interest ... The plume origin on the center line was taken as the closest distance on the center line to the first intercept, and the first intercept was identified as the upstream location for wind on the latitude/longitude plot. [Note: since the plume growth in the power law model is self-similar, the exact position of the plume origin is arbitrary and does not affect the calculation of expansion coefficients $\alpha$, $\beta$ or $\alpha + \beta$.]**

**Section 3.1**

Line 274, "flight tracks shown in Fig. 2" Figure 2 shows time traces of wind speed and wind direction, not flight tracks. The measurements of wind speed and direction seem to have a bias with aircraft flight direction at least this is how I'm interpreting the oscillations shown, especially in panels a) and b). Was this corrected for the plume age calculations? Can vertical lines or another marker be added to show when the plumes were intercepted?

*Response: thank you. We have restated the phrase in the text as:* **...wind speed and direction time series along flight tracks shown in Fig. 2 ...** *Regarding the statement that wind speed and direction may have a bias with aircraft flight direction, this may be true but we have no information to determine what this bias may be or in what aircraft flight direction it occurs. In addition, we observed that the measurements deviate for durations when the aircraft is crossing the plume compared to the durations when the aircraft completes a turn to return to the plume again. This deviation is caused by aerodynamic effects on the AIMMS sensor during the turns. To exclude these deviations from the dataset, we ensured that wind speed and direction corresponding to plume sampling were chosen for analysis only for times when the aircraft was not turning.*

**Section 3.3**

FLEXPART uses a 100 m vertical and horizontal emission source, and the modeled plume structures are presented in Fig. 7. For panels b) and c), the plume height shows very little increase across 15 km but considerable growth of plume width. The measured and modeled boundary

layer heights for those days are higher than the plume heights, implying very slow vertical mixing. How does this compare with von Glasow (2003), who estimate $\beta \simeq 0.6$ for the temperate MBL? FLEXPART seems to show $\alpha >> \beta$ for panels b) and c), but not a). Can the authors discuss this, especially within the context of their observations? If $\alpha >> \beta$ is true for b) and c) but not a), would this have an impact upon the calculation of $\gamma$ in Section 3.5?

*Response: thank you. This is a very informative observation and will help us put our results in broader context. The experiments are only as good as providing an estimate for the combined vertical-horizontal expansion rate, i.e. $\gamma = \alpha + \beta$, because we only have a few plume crossing events for each plume study. In order for us to accurately measure both $\alpha$ and $\beta$ separately, we must have been able to cross the plume more rigorously at multiple horizontal and vertical locations at various downwind distances from the plume source. This was not possible for logistical reasons. Nevertheless, we can observe that the combined vertical-horizontal expansion rate $\gamma$ is less than what was found by von Glasow et al. (2003).*

*We can understand the relative magnitude of vertical versus horizontal mixing (dispersion) by interpreting both Figs. 2 and 7. The study of the vertical and horizontal dispersion of the plume in the FLEXPART-WRF model suggests that, with low wind speeds, suppression of the vertical mixing due to stable conditions is very strong, while there is still horizontal transport and mixing, i.e. $\alpha >> \beta$. With higher wind speeds, the vertical mixing is enhanced, i.e. $\alpha \sim \beta$, but the overall mixing and expansion rate is still lower compared to mid latitude studies. The radiosonde and model results suggest that for all three plumes, the dispersion occurred within the boundary layer and the plume did not reach the top of the boundary layer. Therefore, the method of fitting $\gamma = \alpha + \beta$ is equally valid for all three plumes. We have provided the extra explanation in section 3.5 to make these observation clear:*

**... ($\gamma = \alpha + \beta$ for our case where dispersion occured within the boundary layer and the plumes did not reach the top of the boundary layer) ... An examination of Figs. 2 and 7 suggests that when wind speeds were low (Figs. 2b,c and 7b,c) there was a significant suppression of vertical mixing in comparison to horizontal mixing, i.e. $\alpha >> \beta$, within the stable boundary layer, but when wind speeds were high (Figs. 2a and 7a) vertical mixing was enhanced, i.e. $\alpha \sim \beta$, but the stable boundary layer still resulted in lower overall mixing and expansion rate compared to mid latitude observations.**

In light of the large uncertainty of the fitted line for the first plume intercept, can the authors discuss the uncertainty of the plume age calculation relative to the uncertainty of $CO_2$ enhancement?

*Response: thank you. This is a very interesting point. Turbulence under stable conditions features unique properties that results in intermittent and inhomogeneous mixing of atmospheric constituents, meaning that parcels of air containing fresh $CO_2$ from the plume may experience different rates of dilution as they mix with the atmosphere and disperse. Some parcels may dilute quickly (i.e. to low concentrations) while others may dilute more slowly (remain at high concentrations) as they disperse away from the source. The outcome is a patchy structure for the pollutants. This is why there is a large uncertainty for the fitted line for all plume intercepts, not*

*only because of the uncertainties associated with plume age estimation but also uncertainties with intermittent mixing. In a supporting study for the same campaign by Aliabadi et al. (2016b) we showed that under stable conditions turbulent eddies occur intermittently as large structures with larger atmospheric lifetimes. A Fast Fourier Transform of the turbulent heat flux also showed that energy-containing eddies appear at low frequencies which support this hypothesis. The following statement has been added in section 3.5 to explain this concept:*

**The uncertainties in the expansion rates resulted not only from uncertainties in plume age estimation but also from the intermittent mixing of air parcels under stable conditions (Aliabadi et al., 2016b), which caused nonuniform dilution of $CO_2$ and therefore a scatter in mixing ratios.**

**Section 3.5**

The interpretation of plume expansion rates in the boundary layer (BL) hinges upon whether the plume has mixed vertically to the top of the BL. I may have missed it, but I can find no observations of the extent of plume vertical mixing, but rather a reliance upon FLEXPART. It strikes me as circular logic to use modeled vertical mixing to interpret the observed plumes and thereby calculate mixing parameters that will be used by models. Could the authors use a mass-balance approach (Ryerson et al., 1998; White et al., 1976) to the observed mixing ratios during plume intercepts to evaluate the extent of vertical mixing? Can observed plume widths be compared to the modeled widths, to at least confirm that horizontal mixing is correct in FLEXPART? Can the authors make any calculation of $\alpha$ rather than just $\gamma$?

*Response: thank you for voicing your concern. We disagree with the fact that our logic may be circular. First, in the original Fig. 6 we showed that all pollution peaks from the ship plume were intercepted by the aircraft at an altitude less than 90 m and that no ship pollution peak was intercepted above this height. Second, we use both radiosonde and WRF data to estimate boundary layer height with the method of bulk Richardson number (Aliabadi et al., 2016a) and observe that the estimated measured and modelled boundary layer heights are reasonably close (see Fig. 4). Therefore, based on the evidence, we have a high degree of confidence that the plumes did not rise to the top of the boundary layer. For clarity this discussion has been added to sections 3.1 and 3.3:*

**The boundary layer is statically stable and the boundary layer height is calculated from both radiosonde measurements and the WRF model results, using the method of bulk Richardson number developed by Mahrt (1981) and later used by Aliabadi et al. (2016a). The measurements give boundary layer height as 387 m and 177 m for plumes 2 and 3, while the model gives 350 m, 350 m and 210 m for plumes 1, 2, and 3, respectively. The boundary layer height estimation by the measurements and the model are reasonably close.**

**It is also worth noting that all of the plume crossings that detected ship pollutants above threshold levels occurred below 90 m altitude and no ship pollutants were observed above this height.**

*Thank you for suggesting the references. We studied and learned from them. Again, due to very limited number of intercepts for each plume and variabilities in $CO_2$ concentrations due to patchy structure of pollution dispersion in intermittent turbulence (discussed above), we are not able to estimate plume dispersion height and width, and hence $\alpha$ and $\beta$ separately, from mixing ratios. The best we have been able to do was calculate average emission factors and overall average dispersion rate.*

**Section 3.6**

Figure 8 shows changes in $CO_2$ mixing ratio of approximately 3-7 ppm $CO_2$ for the three plume intercepts shown, but the x-axis of the scatterplots for plume 3 [Fig. 10, panel c) and Fig. 11, panel c)] have a dynamic range of 1.2 ppm. The data shown in Fig. 8 also includes many more data point than is shown in Fig. 10 and 11, panel c). Am I confusing things here? Are the scatterplots (Fig 10 and 11) showing something other than the 1 s data shown in Fig. 8?

*Thenk you. In section 2.2.6 the net peak area method is introduced as a means to calculate emissions factors. In this method every peak is integrated and an average mixing ratio is calculated for the peak to represent excess mixing ratio or concentration versus the background. We use this method to calculate excess mixing ratios in Fig. 10 and 11, while Fig. 8 shows an example time series for a few peaks. From this explanation it is evident that the maximum mixing ratio or concentration for each peak on the time series must be greater than the average value. In section 3.4 we explain that the duration of each peak is identified using a 2 ppb threshold mixing ratio for the $NO_x$ measurement. The opening sentences in sections 3.6.1 and 3.6.2 have been revised to remind the reader of the use of net peak area method in reporting excess mixing ratio or concentrations:*

**Figure 10 shows the scatter plot for excess gas pollutants versus excess carbon dioxide using the net peak area method … Figure 11 shows the scatter plot for excess particle concentrations versus excess carbon dioxide using the net peak area method …**

The fits of $O_3$ vs $CO_2$ have no relevance to either calculations of emission factors or plume dispersion, and should be removed.

*Response: thank you. We have removed the scatter plot and fit for $O_3$ vs. $CO_2$.*

The authors should explain how the uncertainties were calculated for the slopes of the fitted lines presented in Table 1. It appears that the fits are linear regressions, while bivariate least squares fits would be appropriate - both x and y variables have observational errors. Also, the individual uncertainties of the data points is known and can be used to weight the data.

*Response: thank you. We have used a new multivariate least squares approach which takes into account the variability in both x and y components for all linear fits and the calculation of the fit coefficients and the estimation of the associated uncertainties. This approach is called the orthogonal distance regression (ODR). We have added a new section 2.3.1 (Statistical analysis) where we explain this approach and the calculation of the fit coefficients and uncertainties with*

*appropriate references:*

**Regression was required in our analysis to relate one set of measurements to another in order to estimate the plume growth rate and various emission factors. However, since all measurements, including both dependent and independent variables, had inherent uncertainties, the ordinary least squares (OLS) approach could not be used. Instead, a multivariate least squares method, called the orthogonal distance regression (ODR), was used where the sum of squared orthogonal distances between each data point and a linear model was minimized by fitting the model coefficients (Boggs, 1987, 1988). A particular distribution of an ODR algorithm called ODR-PACK was used that took into account the variability of both sample variables, by assuming Gaussian distributions for the uncertainties centered around zero, and fits for the coefficients of the linear model (Boggs, 1989). An error estimate for the coefficients was computed from the linearized quadratic approximation to the chi-squared $\chi^2$ surface at the solution with degrees of freedom $\nu = n - 1$. In addition, a confidence band with 90% probability for the model was computed. A confidence band shows the region within which the model is expected to fall.**

Finally, the stated uncertainty for $CO_2$ is 0.3 ppm, while the range of enhanced $CO_2$ mixing ratios for plume 1 is only 0.2 ppm, i.e. not significant. The data appears to show meaningful correlations, so I suspect the instrumental uncertainties are overly conservative.

*Response: we agree that even though the range for the observed $CO_2$ mixing ratio in plume 1 is in the same order as the reported precision for $CO_2$, it is still possible to find meaningful correlations between $CO_2$ and other pollutant mixing ratios or concentrations. If the precision error in $CO_2$ measurement is random in nature, perhaps with a Gaussian distribution centered at zero, then by merit of multiple measurements one can still observe variations within the precision range for a sample population, but not necessarily any pair of individual measurements. However, the 90% confidence bands for the fitted linear model in plume 1 is considerably larger than plumes 2 and 3 in both figures 10 and 11, indicating that the correlation is poorer for plume 1.*

**Section 3.7**

It appears that Tables 3-7 are redundant with Fig. 12-16. The authors should use only one or the other to present this data. Figures 12-16 would also be well-served to include the uncertainty range for the authors data points, if not for all data.

*Response: thank you. We have improved the quality of Figs. 12-16 by adding the uncertainties for our measurements as error bars, increasing the axes label font size and improving color coding and size for the markers. We have respect for the reviewer's comment realizing that there is some redundancy in Figs. 12-16 and Tables 3-7. However, the figures and tables serve different purposes. On the one hand, the tables provide full numerical data on all of our measurements and reported literature with uncertainties, which are easy to reproduce and can potentially serve future studies. However, tables are difficult to read. On the other hand, the figures communicate the comparison between our measurements and other literature findings effectively by different categorizations using*

*visual means. Therefore, we recommend to keep both, in the interest of a more comprehensive and readable study.*

The $NO_x$ EF reported for plume 1 (open water) appears anomalously low compared to the existing literature. Can the authors discuss whether this is due to their ability to interpret the measured data, or if this is reasonable for the ship under the operating conditions at the time?

*Response: thank you. With the new multivariate orthogonal distance regression analysis, the average emission factor for $NO_x$ in plume 1 has been calculated to be a significantly higher value (almost twice larger than the previous calculation). The new value is now comparable to ships with similar tonnage and ships operated at low engine loads.*

**Technical Corrections**

Figure 15 caption has a typo (for for).

*Response: thank you. This has been corrected.*

Ryerson, T. B., Buhr, M. P., Frost, G. J., Goldan, P. D., Holloway, J. S., Hubler, G., Jobson, B. T., Kuster, W. C., McKeen, S. A., Parrish, D. D., Roberts, J. M., Sueper, D. T., Trainer, M., Williams, J., and Fehsenfeld, F. C.: Emissions lifetimes and ozone formation in power plant plumes, Journal of Geophysical Research-Atmospheres, 103, 22569-22583, 1998.

White, W., Anderson, J., Blumenthal, D., Husar, R., Gillani, N., Husar, J., and Wilson, W.: Formation and transport of secondary air pollutants: ozone and aerosols in the St. Louis urban plume, Science, 194, 187-189, 10.1126/science.959846, 1976.

**Reviewer 2**

**General Remarks**

The manuscript presents results from an in-depth study on emissions from a single icebreaker vessel operating in Arctic waters. Although there is already extensive information available on the gaseous and particulate emissions from marine Diesel engines, this study adds significant data on ship emissions in the Arctic. The study is well designed and carefully conducted. It covers both the experimental determination of emission factors and the investigation of ship emissions transport in the marine boundary layer under the specific conditions of Arctic waters. The manuscript is in the scope of the journal and the presentation of the material is clear and well structured. Few minor revisions should be considered before publication in ACP. Technically, most of the figures require careful revision because they are hard to read in the current version.

*Response: thank you. We have paid great attention to improve the figure qualities. We have increased font sizes on axes labels and legends, used larger markers sizes, increased line widths, used a diverse color palette, and reduced clutter to communicate visual results more effectively.*

**Specific Comments**

**1**

The only major comment concerns the determination of emission factors from the plume encounters. In the present form, the authors report average emission factors for each investigated species. However, this way of presenting the data neglects potential effects of transformation of the species during plume expansion and atmospheric transformation. Combining the information shown in Fig. 9, 10, and 11, it becomes evident that plume encounters were measured at plume ages between 100 and > 1000 s. This span of plume ages should allow the investigation of emission factors at various plume ages, i.e., the determination of effective emission factors. It is recommended to determine emission factors separately for all plume encounters and plot the data as a function of plume age. Doing this would permit investigating whether there was chemical or physical transformation of the investigated species during plume ageing; see for example the results on particle ageing shown in Petzold et al. (2008) and Tian et al. (2014).

*Response: thank you. We are in agreement with the reviewer that it would have been very informative to study emission factors and processing as a function of plume age due to the fact that the pollutants would experience physical and chemical processing in the marine boundary layer over time. We have previously studied references by Petzold et al. (2008) and Tian et al. (2014) and found them excellent articles. To do the detailed analysis it would have been necessary to sample the plume for long durations and numerous intercepts at various distances downwind of the emission source. However, for logistical reasons, this was not possible. Instead the plume was sampled for short durations (on average 5.8, 20.2, and 38.2 s for plumes 1, 2, and 3, respectively) and for very limited number of intercepts (11, 12, and 29 intercepts for plumes 1, 2, and 3,*

*respectively). As a result, a study of the emission factor variation and pollutant processing over the marine boundary layer as a function of plume age could not be performed in a statistically significant way. In fact we attempted to study the evolution of particle size distribution given by an SMPS instrument to understand particle growth behavior as a function of plume age. However, the SMPS would have required a minimum of 1-minute sampling of the plume to give meaningful size distributions, which was not a possibility for our study. Therefore, we only report average emission factors for the entire plume, which we have calculated in a statistically significant way. This was already stated at the end of section 2.2.6:*

**Due to limited number of plume intercepts in this study, we compute average emissions factors for all plume intercepts.**

**2**

In the section on applied particle instruments, the information on lower detection sizes is missing. Reporting aerosol data as CPC, OPC or UHSAS is misleading. Instead they should be reported as, e.g., N(OPC, Dp 250 nm) etc. Then the reader recognizes directly the aerosol mode covered by the respective instrument. Furthermore, minimum detection diameters of the instruments should be added to Section 2.2.3.

*Response: thank you for this important suggestion. We have referred to particle number concentrations using the symbols $N_{OPC}$, $N_{CPC}$, etc. where the subscript shows the instrument with which the number concentration was measured. We have revisited section 2.2.3 and made sure that the minimum detection diameters for all relevant instruments are mentioned. The following text is added to rBC measurements:*

**The instrument used a continuous intra-cavity laser to classify aerosol particles as either rBC containing or purely scattering. While non-rBC particles only scattered the laser light at the same wavelength, an absorptive rBC-containing particle passing through the laser beam was heated to its incandescence temperature and the resulting thermal radiation was measured by optical detectors as the particle vaporized. The peak incandescence signal was linearly related to the rBC mass. Size selected Fullerene Soot particles were used as calibration standard. The detection efficiency of this SP2 (version D) drops off for particles smaller than 80 nm.**

**3**

Careful revision of all figures is strongly recommended. They should be made much simpler and need to be enlarged. Figures 12 to 16 are not really necessary since the information is already contained in the extensive tables.

*Response: thank you. We have revisited all Figures. We have reduced clutter, used larger fonts on axis labels, increased marker sizes and line widths, used larger fonts for legends, and improved the color choices to communicate results most effectively. We respect the reviewers comment on*

*the redundancy of Figs. 12-16 and Tables 3-7. However, the figures and tables serve different purposes. On the one hand, the tables provide full numerical data on all of our measurements and reported literature with uncertainties, which are easy to reproduce and can potentially serve future studies. However, tables are difficult to read. On the other hand, the figures communicate the comparison between our measurements and other literature findings effectively by different categorizations using visual means. Therefore, we recommend to keep both, in the interest of a more comprehensive and readable study.*

The presentation of emission factors in Tables 3 to 6 should focus on the comparison of reported values with data for similar engine types. Currently, the tables list all values available in literature without giving information whether or not the investigated engines are comparable to the engines operated aboard the Amundsen. Giving more weight to those engines of similar types would increase the readability of Tables 3 to 6 significantly.

*Thank you. Tables 3-6 merely summarize all numerical values and uncertainties for future reproduction of our results or other studies. For the purpose of comparison of the emission factors we calculate with other studies, we make the best use of Figs. 12-16. Here the reader can compare the computed value against similar tonnage ships in the literature in the middle panel for each figure. This would correspond to High Speed Diesel (HSD) and Medium Speed Diesel (MSD) categories. The reader can also compare our emission factors with different engine loadings on the third panel, which would correspond to low loading conditions in other studies. The reader can also compare our results with engine type, or equivalently the fuel type they consume by studying the legends with Marine Gas Oil (MGO). With this discussion, it is apparent that Figs. 12-16 are very useful.*

**Minor Remarks**

Page 1, line 7: suggested rephrasing: "Canadian Coast Guard icebreaker Amundsen".

*Response: thank you. This has been implemented.*

Page 6, line 169: change to (OPC GRIMM Model 1.129).

*Response: thank you. This has been implemented.*

**References**

Petzold, A., Hasselbach, J., Lauer, P., Baumann, R., Franke, K., Gurk, C., Schlager, H., and Weingartner, E.: Experimental studies on particle emissions from cruising ship, their characteristic

properties, transformation and atmospheric lifetime in the marine boundary layer, Atmos. Chem. Phys., 8, 2387-2403, 2008.

Tian, J., Riemer, N., West, M., Pfaffenberger, L., Schlager, H., and Petzold, A.: Mod- eling the evolution of aerosol particles in a ship plume using PartMC-MOSAIC, 14, 5327-5347, doi: 10.5194/acp-14-5327-2014, 2014.

---

## Author Response (AR2)

**Ship emissions measurement in the Arctic by plume intercepts of the Canadian Coast Guard icebreaker *Amundsen* from the *Polar 6* aircraft platform**

By A. A. Aliabadi* et al.

*Department of Architecture, Massachusetts Institute of Technology, Cambridge, USA

Response to reviewers

Manuscript ID: acp-2015-1032

June 10, 2016

Dear Dr. Harald Saathoff

We thank you for your time and the second evaluation of the manuscript acp-2015-1032. Based on your comment we have extended the captions for Figs. 13-16 and Tabels 3-7 so that they are now self-explanatory. We have uploaded all necessary files for production in the portal, including a complete version of the manuscript in PDF format, the manuscript source file in TeXformat, the references in bibliography in TeXformat, and all figures in zipped format. We hope that this submission will assist your final evaluation of the manuscript and initiate the production phase. Please do not hesitate to ask us whether you need any further information.

Best Regards,

Amir A. Aliabadi, on behalf of all co-authors